# Development of a sensitive real-time quaking-induced conversion (RT-QuIC) assay for application in prion-infected blood

**Charlotte M. Thomas**\*, **M. Khalid F. Salamat**, **Christopher de Wolf**, **Sandra McCutcheon, A. Richard Alejo Blanco, Jean C. Manson, Nora Hunter, E. Fiona Houston**

The Roslin Institute, Royal (Dick) School of Veterinary Studies, University of Edinburgh, Easter Bush Campus, Midlothian, United Kingdom

\* Charlotte.Thomas@ed.ac.uk

## Abstract

Efforts to prevent human-to-human transmission of variant Creutzfeldt-Jakob disease (vCJD) by contaminated blood would be aided by the development of a sensitive diagnostic test that could be routinely used to screen blood donations. As blood samples from vCJD patients are extremely rare, here we describe the optimisation of real-time quaking-induced conversion (RT-QuIC) for detection of PrP$^{Sc}$ (misfolded prion protein, a marker of prion infection) in blood samples from an established large animal model of vCJD, sheep experimentally infected with bovine spongiform encephalopathy (BSE). Comparative endpoint titration experiments with RT-QuIC, miniaturized bead protein misfolding cyclic amplification (mb-PMCA) and intracerebral inoculation of a transgenic mouse line expressing sheep PrP (tgOvARQ), demonstrated highly sensitive detection of PrP$^{Sc}$ by RT-QuIC in a reference sheep brain homogenate. Upon addition of a capture step with iron oxide beads, the RT-QuIC assay was able to detect PrP$^{Sc}$ in whole blood samples from BSE-infected sheep up to two years before disease onset. Both RT-QuIC and mb-PMCA also demonstrated sensitive detection of PrP$^{Sc}$ in a reference vCJD-infected human brain homogenate, suggesting that either assay may be suitable for application to human blood samples. Our results support the further development and evaluation of RT-QuIC as a diagnostic or screening test for vCJD.

## Introduction

Prion diseases are fatal neurodegenerative disorders that affect both humans and animals. They include scrapie in sheep/goats, chronic wasting disease (CWD) in cervids, bovine spongiform encephalopathy (BSE) in cattle, and Creutzfeldt-Jakob disease (CJD) in humans [1,2]. A key feature of prion diseases is the accumulation of a misfolded, aggregated form of the prion protein (designated PrP$^{Sc}$) in the brain (and, in some instances, the secondary lymphoid tissues) [3–6]. PrP$^{Sc}$ self-propagates by inducing the normal, cellular form of the prion protein

**Funding:** The results presented in this paper are based on independent research commissioned and funded by the Policy Research Programme of the Department of Health and Social Care (https://www.nihr.ac.uk/explore-nihr/funding-programmes/policy-research.htm). The award ("Comparative evaluation of the performance of proposed diagnostic tests for vCJD in preclinical blood samples"; NIHR reference: PR-R17-0916-23006) was made to EFH. The views expressed in the publication are those of the authors and not necessarily those of the Department of Health and Social Care, "arms" length bodies or other government departments. The funders had no role in study design, data collection and analysis, decision to publish, or preparation of the manuscript.

**Competing interests:** The authors have declared that no competing interests exist.

(termed PrP or PrP$^C$) to misfold into a likeness of itself [7]. The misfolded isoform is insoluble, with an increased tendency to aggregate, initiating a misfolding cascade that results in deposition of PrP$^{Sc}$ aggregates and fibrils, with associated neurotoxicity. Thus, PrP$^{Sc}$ functions as the infectious agent (or, "prion") responsible for disease transmission/pathology [8], and as a biomarker for prion infection.

Human prion diseases typically have a sporadic or genetic aetiology [9], but they can also be acquired by infection [1]. Acquired prion diseases occur through iatrogenic exposure to infected tissues/body fluids (by surgical procedures, blood transfusions), or, occasionally, from dietary exposure to prion-infected tissues or animal products. Variant Creutzfeldt-Jakob disease (vCJD), a novel human prion disease that emerged in the UK during the 1990s [10], was caused by eating meat from cattle infected with BSE [11–14]. The causal link between BSE (also known as "mad cow disease") and vCJD was confirmed by mouse transmission studies, which demonstrated that a single prion agent (or, "strain") was responsible for the two diseases [11,13].

A total of 232 clinical cases of definite or probable vCJD have been reported (to date), including 178 cases in the UK [15]. While dietary exposure to BSE represents the predominant transmission route for vCJD [14], several cases have been attributed to blood transfusions with contaminated blood products from donors who were asymptomatic at the time of blood donation, but later went on to develop clinical signs of vCJD [16–18]. Although the incidence of vCJD is now low (the last UK case was reported in 2016 and there have been no new transfusion-related cases since 2006) [15], concerns about the risk of secondary transmission of vCJD persist. These stem from large-scale surveys of archived appendix tissue, removed during routine appendectomy, which detected abnormal PrP deposition (a marker for vCJD infection) in a small number of samples. Extrapolation of the data suggests that up to 1 in 2000 people in the UK could be harbouring silent vCJD infections [19–21]. As prion diseases often exhibit long asymptomatic incubation periods (ranging from several years to decades) [22], it is uncertain if these individuals are preclinically infected (will go on to develop vCJD) or subclinically infected (will not develop vCJD during their natural lifespan). Their potential to transmit infection by blood transfusion is also uncertain.

A validated blood test for vCJD that could accurately detect "silent" (subclinical or preclinical) infections would be useful to refine current prevalence estimates and to improve early diagnosis. Such a test could also be used to screen blood donations, which would potentially be invaluable for preventing further spread of the disease. However, the development of such a blood test has been challenging; partly because the concentration of the infectious agent (PrP$^{Sc}$) in body fluids is very low in comparison to tissues [23,24], with levels during the preclinical stage of infection predicted to be lower still [25].

Recently, several *in vitro* prion assays have been developed that address this issue by amplifying low levels of PrP$^{Sc}$ present in biological samples; these include protein misfolding cyclic amplification (PMCA) and real-time quaking-induced conversion (RT-QuIC), both of which exploit the ability of PrP$^{Sc}$ to self-propagate. In PMCA, the PrP$^{Sc}$ in an infected tissue or blood sample acts as a template (or "seed") to induce misfolding of the normal PrP$^C$ "substrate" in uninfected brain tissue [26,27]. Amplification proceeds through cycles of incubation and sonication (to break up PrP$^{Sc}$ aggregates and thereby increase the number of nuclei for seeded conversion) until PrP$^{Sc}$ levels are sufficiently high for detection by Western blot [28]. PMCA was first shown to detect PrP$^{Sc}$ in preclinical and clinical blood samples from scrapie-infected hamsters [29,30]. These results have since been replicated in other animal models of prion disease, including sheep infected with BSE [31,32] and primates infected with vCJD [31,33]. In two independent blinded studies on blood samples from vCJD patients, different versions of PMCA identified blood samples from vCJD patients with 100% sensitivity and 100% specificity [34,35], and one was also able to detect PrP$^{Sc}$ in a small number of preclinical samples [34].

RT-QuIC, although based on the same principle as PMCA, has several technical differences that may make the assay more amenable to high-throughput screening systems [36]. RT-QuIC uses bacterially expressed recombinant PrP (recPrP) as the PrP$^C$ substrate, a more humane and cost-effective alternative to the rodent brain homogenates commonly used in PMCA, which can be produced in larger quantities with better quality control [37,38]. RT-QuIC uses intermittent shaking in a plate reader to generate nuclei for seeded conversion, which is easier to standardise and more reproducible than sonication [37,38]. Furthermore, the assay measures PrP aggregation in real time through the binding of Thioflavin T (ThT), producing a fluorescent signal that is amenable to quantitative analysis [36].

RT-QuIC is now routinely used to test patient cerebrospinal fluid (CSF) samples for sporadic CJD [39], but is less effective at detecting PrP$^{Sc}$ in more complex biological samples, such as blood [40,41]. Different pre-amplification sample processing strategies, including sodium phosphotungstic acid (NaPTA) precipitation and iron oxide magnetic extraction (IOME), have been shown to aid RT-QuIC detection of PrP$^{Sc}$ in blood and other body fluids/excreta of deer infected with CWD [42–47]. By incorporation of an immunoprecipitation step, a modified version of RT-QuIC, termed "enhanced QuIC" (eQuIC), was shown to enhance detection of vCJD brain tissue spiked into human plasma [40], and detect low levels of PrP$^{Sc}$ in serum and plasma samples from scrapie infected rodents [40,48]. However, there is no published data on the application of RT-QuIC to endogenously-infected blood samples from vCJD patients, or its application to blood samples from a large animal model of vCJD.

As the number of blood samples available from presymptomatic vCJD cases is limited, animal models offer the only feasible alternative source of preclinical samples for development and initial evaluation of blood tests for vCJD. Using blood samples archived from a previous study using sheep experimentally infected with BSE to study the transmission of prion diseases by blood transfusion (an established large animal model of vCJD) [32,49], we have adapted and optimised RT-QuIC for detection of BSE in sheep blood.

Herein, we compare the analytical sensitivity of RT-QuIC with that of an established miniaturized bead-PMCA (mb-PMCA) assay [50], and infectivity titres estimated by bioassay in a transgenic mouse line (tgOvARQ) expressing sheep PrP [51]. By addition of an iron oxide magnetic extraction (IOME) pre-amplification step, we describe a modified version of RT-QuIC, termed "whole blood IOME RT-QuIC" (WB IOME RT-QuIC), which is shown to detect PrP$^{Sc}$ in endogenously infected blood samples from BSE-infected sheep at both clinical and preclinical stages of infection. Furthermore, we demonstrate sensitive detection of PrP$^{Sc}$ in a reference vCJD brain homogenate by both RT-QuIC and mb-PMCA, suggesting applicability of both methods to human samples. The outcomes of this study support the further development and evaluation of WB IOME RT-QuIC as a potential blood-based diagnostic or screening test for vCJD.

## Materials and methods

### Ethics statement

All animal work was reviewed and approved by Animal Welfare and Ethical Review Bodies at The Institute for Animal Health, United Kingdom and The Roslin Institute, and conducted under the authority of Home Office Project Licences (PPL 30/2282; 60/4143; 70/8595), and in accordance with ARRIVE guidelines. Appropriate care was provided to minimise harm and suffering, and anesthesia was administered where necessary. Human brain tissue was sourced through the MRC Edinburgh Brain Bank (Edinburgh, Scotland, UK) (East of Scotland Research Ethics Service, Ref 21/ES/0087). Human vCJD brain tissue was sourced through the National Institute for Biological Standards and Controls (NIBSC), Medicines and Healthcare

products Regulatory Agency (MHRA) (South Mimms, England, UK) (East Midlands - Derby Research Ethics Committee, Ref 19/EM/0314).

## Reference BSE-infected sheep and vCJD-infected human brain homogenates

A reference BSE-infected sheep brain homogenate pool (reference BSB/7/10), consisting of brain tissue samples pooled from five individual sheep (*PRNP* genotype ARQ/ARQ) clinically affected with BSE following experimental oral infection with 5 g BSE-infected cattle brain homogenate, was prepared as a 10% (w/v) homogenate in PBS (pH 7.4). Uninfected, negative control ovine brain tissue was collected from individual ARQ/ARQ sheep that had been mock-infected with 5 g uninfected bovine brain homogenate [32,49].

Reference variant CJD-infected human brain tissue was supplied by NIBSC as a 10% (w/v) brain homogenate in PBS (pH 7.4) (reference NHBY0/0003; codon 129 MM). Negative control (sudden death) human frontal cortex brain tissue, was provided by the Medical Research Council (MRC) Edinburgh Brain and Tissue Bank (reference BBN001.34150) and prepared as a 10% (w/v) brain homogenate in PBS (pH 7.4).

## Sheep experiments (source of blood samples)

The blood samples used in this study were archived during a previous study to determine the effectiveness of leucodepletion in removing prion infectivity from sheep blood components relevant to those used in human blood transfusion [32,49]. All sheep had *PRNP* genotype ARQ/ARQ (amino acids encoded by codons 136, 154 and 171 of the ovine *PRNP* gene), which confers high susceptibility to BSE [52]. Donor sheep were orally infected with 5 g BSE-infected cattle brain homogenate, and blood was collected at 10 months post infection (mpi) for preparation of blood components, which were transfused intravenously into individual recipient sheep. Recipient sheep received sedation/analgesia by injection of medetomidine and butorphanol during placement of the intravenous catheter. Donor and recipient sheep were monitored regularly for clinical signs of BSE, or other conditions that might compromise their welfare, and sacrificed by intravenous injection of pentobarbital sodium solution when humane end points were reached. Efforts to alleviate suffering included housing the sheep in small groups throughout the experiment to alleviate social stress and using a standardized scoring system for assessing the severity of clinical signs, specified in the Home Office licence, with pre-defined criteria for determining humane end points. BSE infection status of each sheep was confirmed by detection of PrP$^{Sc}$ by Western blot and immunohistochemistry (IHC) in brain and lymphoid tissue samples collected at necropsy. Additional blood samples (in EDTA anticoagulant) were collected from selected donor and recipient sheep before infection and at regular intervals post-infection until they were culled. These blood samples were separated into fractions (plasma, red cells, buffy coat) as previously described [32,49], and multiple aliquots of whole blood and blood fractions were stored at -80˚C.

## Mouse experiments (tgOvARQ bioassay)

Transgenic Tg(OvPrP-A136)3533 mice expressing the ovine PrP gene (*PRNP* allele ARQ) were generated by Glenn Telling (Colorado State University), as previously described [51], and transported to the Roslin Institute, Edinburgh, where they were crossed onto a *Prnp*$^{0/0}$ 129/Ola genetic background [53]. Groups of the resulting hemizygous (PrP+/-) transgenic mice (hereafter referred to as tgOvARQ mice) (mixed-sex; 2–4 months of age) were each inoculated intracerebrally (20 µl/mouse) with a ten-fold dilution series ($10^{-1}$ to $10^{-8}$) of the reference BSE-infected sheep brain homogenate pool (BSB/7/10). Intracerebral inoculations were

performed under general anaesthesia induced and maintained by administration of vaporized isoflurane. Mice were housed in a Biological Containment Level 3 (BCL3) animal facility at the Roslin Institute, and regularly monitored for the development of clinical signs of BSE, or other conditions that might compromise their welfare, up to 800 days post-infection, and sacrificed by cervical dislocation when humane end points were reached. Efforts to alleviate suffering included screening of inoculum for bacterial contamination prior to inoculation, to minimise the risk of adverse reactions, and using a standardized scoring system for assessing the severity of clinical signs, specified in the Home Office licence, with pre-defined criteria for determining humane end points. At necropsy, one half of the brain and spleen were flash frozen in liquid nitrogen and the other half fixed in 10% formal saline for 72 hours before embedding in paraffin wax. Frozen tissues were stored at -80˚C until use. BSE infection status was confirmed by detection of PrP$^{Sc}$ by Western blot and IHC in brain samples collected at necropsy.

### Miniaturized bead-PMCA (mb-PMCA assay)

The miniaturized bead-PMCA (mb-PMCA) protocol was developed and published previously [31,50]. Brains from healthy TgShpXI mice (over-expressing sheep PrP$^{C}$; *PRNP* genotype ARQ) [54] were used as a source of PrP$^{C}$ ("substrate") for PMCA. TgShpXI mouse brains, provided by Dr Olivier Andreoletti (ENV, INRA, Toulouse, France), were prepared as 10% (w/v) homogenates in pre-chilled PMCA buffer (PBS (pH 7.2), 150 mM NaCl, 0.25% (v/v) Triton X-100, supplemented with one tablet of EDTA-free complete protease inhibitor (Roche) per 50 ml buffer). 10% (w/v) reference brain homogenates prepared in PBS (pH 7.4) were serially diluted in the same PMCA buffer.

PMCA reactions were performed in 96 well PCR microtiter plates (Axygen), with each well containing 45 µl of substrate and 1 Teflon bead (2.381 mm diameter; Marteau & Lemarié, Inc.). Reactions were seeded with 5 µl brain homogenate or buffy coat sample, and dextran sulphate solution (Sigma-Aldrich) was added to a final concentration of 0.5% (w/v). Reactions were run in a microplate horn attached to a programmable Misonix Q700 sonicator (QSonica) with a water recirculation system through the horn to maintain temperature at 37˚C, as previously described [32]. The amplitude of the sonicator was set at 50–65% to maintain the minimum energy output at 3500 W per cycle, and each round of amplification consisted of 96 cycles of 10 sec sonication, 14 min 50 sec incubation at 37˚C. After one round of amplification, reaction products were diluted 1:10 with fresh substrate to seed the following round. Following the second round of PMCA, reaction products (18 µl) were digested with proteinase-K (0.1 mg/ml) at 37˚C. PK-digested products (3.6 µl) were resolved by SDS-PAGE (NuPAGE™, ThermoFisher Scientific) before transfer onto a nitrocellulose membrane. Western blotting for detection of PrP$^{Sc}$ was performed using 0.25 µg/ml mouse monoclonal antibody ROS-BC6 (anti-sheep PrP; epitope amino acids 144–154), as previously described [32].

### Recombinant prion protein (recPrP) expression and purification

The recombinant PrP$^{C}$ substrate (recPrP) used for RT-QuIC experiments was N-terminally truncated ovine recPrP, spanning amino acid residues 94–233 (sheep ARQ allele; RefSeq: NP_001009481.1; UniProt: Q7JK02.1). Purification of recPrP was performed as previously described [36], with modifications. Plasmid DNA containing the sequence for truncated ovine recPrP (pTrc vector, ThermoFisher Scientific) was transformed into *E. coli* Rosetta ™ 2(DE3) competent cells (Merck). Cells were grown in terrific broth (Sigma-Aldrich) supplemented with 100 µg ml$^{-1}$ ampicillin and 34 µg ml$^{-1}$ chloramphenicol, and expression of recPrP induced using the Overnight Express™ Autoinduction system (Novagen), according to manufacturer's instructions.

Cell pellets, harvested from 1 L bacterial culture, were subjected to two cycles of freeze thaw at -80°C, before lysis with BugBuster Master Mix™ reagent (Novagen) to isolate inclusion bodies. Inclusion bodies were denatured in 8 M guanidine-HCl (pH 8) and centrifuged at 11,000 x g for 10 min. The supernatant was mixed with 50 ml Ni-NTA Superflow resin (Qiagen) equilibrated in denaturing buffer (100 mM sodium phosphate (pH 8), 10 mM tris-base, 6 M guanidine-HCl), and the resulting resin-recPrP mixture loaded onto an XK chromatography column (GE Healthcare), to be purified using an AKTA Explorer FPLC (GE Healthcare). RecPrP was refolded with a gradient of denaturing to refolding buffer (100 mM sodium phosphate (pH 8), 10 mM tris-base) over 200 minutes, and eluted from the column with a gradient of refolding to elution buffer (refolding buffer supplemented with 550 mM imidazole) over 60 minutes. Eluted recPrP was collected at a 1:1 volume ratio into chilled dialysis buffer (10 mM sodium phosphate (pH 6.5)).

The purification process was monitored by UV. Protein fractions above 0.4 absorbance units (AU) were pooled, filtered with a 0.22 μm syringe filter, and dialysed in 5 L dialysis buffer (4°C, static) for approximately 20–22 h (including one change of dialysis buffer). Dialysed recPrP was then re-filtered, aliquoted, and stored at -80°C. Protein concentration was determined by measuring absorbance at 280 nm ($A_{280\ nm}$), and purity estimated by SDS-PAGE (NuPAGE™, ThermoFisher Scientific) followed by Coomassie blue staining (Expedeon Instant-Blue™ Abcam, UK) or silver staining (SilverQuest ™ ThermoFisher Scientific), according to manufacturer's instructions.

## Real-Time Quaking-Induced Conversion (RT-QuIC)

**RT-QuIC seeded with brain homogenates.** 10% (w/v) reference brain homogenates prepared in PBS (pH 7.4) were serially diluted in PBS supplemented with 0.025% SDS and 1% (v/v) N2 supplement (100X) (Gibco), before addition to RT-QuIC reactions. RT-QuIC was carried out as previously described [36], with modifications. Briefly, aliquots (98 μl) of RT-QuIC reaction buffer (PBS (pH 7.4), 170 mM NaCl, 1 mM EDTA, 10 μM Thioflavin T (ThT)) containing 0.1 mg/ml recPrP which had filtered through a 100 kDa MWCO spin filter (Nanosep ™, Pall) were added to the wells of a black, clear-bottomed 96-well plate (Nunc). Individual wells were then "seeded" with 2 μl of the appropriate brain homogenate sample. Plates were sealed with sealing film (Sigma-Aldrich) and incubated at 50°C in a POLARstar Omega plate reader (BMG Labtech) with 1 min cycles of shaking (700 rpm, double orbital) followed by 1 min rest. ThT fluorescence was recorded every 15 min (450 nm excitation, 480 nm emission, gain 2000) for $\geq$ 50 h.

**RT-QuIC seeded with spiked blood.** In assays of spiked blood, a spike of 10% (w/v) BSE-infected sheep brain homogenate (from a single sheep) was diluted 1:10 in whole blood from an uninfected sheep. Serial (ten-fold) dilutions of this initial spiked sample were also prepared in uninfected blood. For comparison, the same BSE-infected sheep brain homogenate was spiked into PBS, rather than whole blood (with subsequent dilutions also prepared in PBS). Negative controls included a ten-fold dilution series of uninfected sheep brain homogenate spiked into uninfected sheep blood. In initial experiments, spiked blood samples (2 μl) were added directly to RT-QuIC reaction mix (98 μl). In subsequent experiments, spiked blood samples (5 μl) were diluted in 1 ml PBS buffer (PBS (pH 7.4), 1mM MgCl$_2$, 250 U/ml benzonase (> 99% purity)), before treatment with iron oxide beads (see next section).

**RT-QuIC seeded with BSE-infected sheep blood.** For whole blood IOME RT-QuIC (WB IOME RT-QuIC), samples of whole sheep blood (200 μl) were thawed at ambient temperature and added to tubes containing 1.3 ml "IOME capture buffer" (100 mM Tris-HCl (pH 8.4), 0.1% w/v CHAPS, 1% w/v BSA fraction V, 10 U/ml benzonase (>99% purity),

supplemented with one tablet of EDTA-free complete protease inhibitor (Roche) per 50 ml buffer) and 4 μl iron oxide beads (Bangs Laboratories, Inc.), which had been washed three times in 1 ml PBS (pH 7.4). Blood-bead mixtures were then incubated at ambient temperature on an end-over-end rotator overnight. The iron oxide beads were isolated on a DynaMag magnet (Invitrogen) and washed twice with PBS. Bead mixtures were sonicated for 1 minute at 50% amplitude in 1 ml PBS using a Branson 450 Digital Sonifier, before two additional wash steps with PBS and another round of sonication (1 minute at 50% amplitude). The beads were then magnetically separated and resuspended in 20 μl PBS-SDS (PBS (pH 7.4) supplemented with 0.025% SDS). Resuspended beads (2 μl) were used to seed 98 μl RT-QuIC reaction buffer, and the RT-QuIC reaction performed using the same cycling parameters described for RT-QuIC seeded with brain homogenates.

**RT-QuIC analytical parameters.** RT-QuIC experiments seeded with brain homogenate were assigned a cut-off time of 50 h, whereas WB IOME RT-QuIC experiments were assigned a cut-off time of 20 h. Cut-off times were chosen based on optimisation experiments which demonstrated a substantial increase in the rate of false positives when the respective assay was run for longer. Samples were considered positive if 50% or more replicate reactions exceeded threshold fluorescence within the designated cut-off time. For $SD_{50}$ experiments, the time taken for an individual reaction to exceed threshold was defined as the "lag time" ($t$). The inverse of this value ($1/t$) was deemed equivalent to the "amyloid formation rate" and was used to infer RT-QuIC seeding activity [46]. Reactions that did not exceed threshold fluorescence within the timeframe of the experiment were assigned an amyloid formation rate of zero.

For RT-QuIC experiments seeded with brain homogenate, threshold was set at 200% of the mean fluorescence reading for all negative control measurements up to the designated cut-off time [36]. As WB IOME RT-QuIC experiments had a higher rate of unseeded fibrilisation, a more stringent threshold value was chosen; here, threshold fluorescence was set at 75% of the maximum fluorescence reading for the plate (75% $F_{max}$, equivalent to 195,000 RFU). This value was chosen based on Receiver Operator Characteristic (ROC) analysis of WB IOME RT-QuIC optimisation data. For ROC analysis, raw fluorescence data was normalised to % fluorescence relative to $F_{max}$ (% $F_{max}$) and the theoretical sensitivity and specificity of the assay was calculated at a range of putative threshold values. These data were plotted as fractions (range 0–1), and the area under the curve (AUC) metric was used to describe the overall predictive ability of the assay [55].

## Data analysis

**$SD_{50}$ and $ID_{50}$ calculations.** $SD_{50}$ values, representing the seeding dose where $\geq$ 50% replicate reactions score positive for seeding activity (RT-QuIC) or PK-resistant $PrP^{Sc}$ (PMCA), were calculated according to the Spearman-Kärber Eq (1): where $X_{p=1}$ represents the highest log dilution giving all (10/10) positive responses; $d$, the log dilution factor; $p$, the proportion of positive responses at a given brain dilution; and $d\Sigma p$, the sum of values of $p$ for $X_{p=1}$ and all higher dilutions (to the most dilute sample tested [56,57].

$$\text{Log}_{10}\ SD_{50} = X_p{=}_1 + 1/2d - d\sum p \tag{1}$$

$$\left[\sum \left(p(1-p)/n-1\right)\right]^{1/2} \tag{2}$$

$\text{Log}_{10}$ $SD_{50}$ values were adjusted to report the number of $SD_{50}$ units per gram of brain tissue, and standard error of the mean (SEM) calculated according to Eq (2): where $n$ is the number of replicates [56,57]. $ID_{50}$ values (the infectious dose where $\geq$ 50% of inoculated mice

became infected) were calculated using the same methodology. Fold-changes in sensitivity were calculated by comparing mean $SD_{50}$ or $ID_{50}$ values/g brain tissue (according to data in S1–S3 Tables).

**Statistical analysis of RT-QuIC amyloid formation rates.** RT-QuIC amyloid formation rates were compared using a nonparametric Wilcoxon signed-rank test in GraphPad Prism v9.2 (GraphPad Software Inc., CA, USA), with $p$-values $< 0.05$ considered significant. Median amyloid formation rates were compared against a theoretical median of zero (equivalent to zero amplification in negative control datasets). If a dataset had a median value matching the hypothetical value, that dataset was omitted from analyses. The rationale for choosing this statistical model was as follows: (i) our sample size was typically small (10–14 replicates), and data were often right skewed due to multiple zero readings at higher brain homogenate dilutions. Therefore we could not assume a normal distribution; (ii) in some cases, all values for a given sample were equal making other nonparametric rank-based tests, such as the Mann-Whitney test, invalid (as described by [58,59]).

# Results

## Optimisation of RT-QuIC for detection of seeding activity in BSE-infected sheep brain

To adapt RT-QuIC for detection of prions in BSE-infected sheep tissues/biological samples, initial experiments were conducted to determine the optimal conditions for amplification of $PrP^{Sc}$ (misfolded PrP, a marker for prion infection) from BSE-infected sheep brain homogenate, including which recombinant $PrP^C$ (recPrP) substrate to use. Various recPrP substrates were tested, including hamster-sheep chimeric recPrP previously used for the eQuIC assay [40]. After extensive optimisation (summarised in S1 Fig), N-terminally truncated ovine (sheep) recPrP (amino acids 94–233; *PRNP* genotype ARQ) demonstrated the most favourable RT-QuIC metrics (shortest lag time, highest analytical sensitivity), and was therefore selected as the substrate. Truncated ovine recPrP was readily expressed in, and purified from, Rosetta (DE3) *E. coli* cells, resulting in high purity preparations of monomeric recPrP, as assessed by silver staining (S2 Fig), with typical yields ranging between 20–50 mg purified protein per litre of bacterial culture.

Since recPrP may undergo unseeded (spontaneous) fibrilisation during RT-QuIC [36,60] it was important to define the analytical parameters of the assay to ensure optimal discrimination between true and false positive readouts. The cut-off time for analysing data from endpoint titration experiments was set at 50 h, because minimal sensitivity was gained beyond this time, and because the rate of unseeded fibrilisation increased substantially when the reaction was run for longer (S3 Fig). Positive replicates were defined as reactions in which ThT fluorescence exceeded a pre-defined threshold value (equivalent to twice the mean fluorescence for all negative control reactions on the plate) within the first 50 hours of the experiment.

According to these analytical parameters, RT-QuIC consistently detected seeding activity (indicating the presence of $PrP^{Sc}$, a subset of which "seeds" amyloid formation) in $10^{-4}$ to $10^{-7}$ dilutions of a reference brain homogenate pool from BSE-positive sheep (reference code: BSB/ 7/10) (Fig 1A). Amyloid formation rates at these dilutions ($10^{-4}$ to $10^{-7}$) were significantly different to negative controls (i.e. reactions seeded with $10^{-4}$ to $10^{-10}$ dilutions of brain tissue from mock-infected negative control sheep) ($p < 0.05$, Wilcoxon signed-rank test), whereas amyloid formation rates at higher dilutions ($10^{-8}$ to $10^{-10}$) were not significantly different to negative controls (Fig 2A). Although amyloid seeding activity was detected to an endpoint dilution of $10^{-8}$, only 1/10 replicate reactions tested positive at this dilution (Figs 1A and 2A),

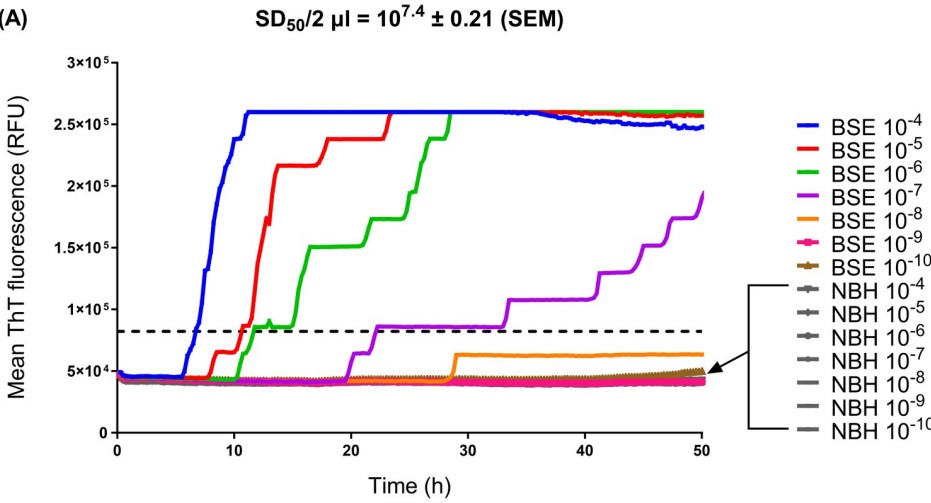

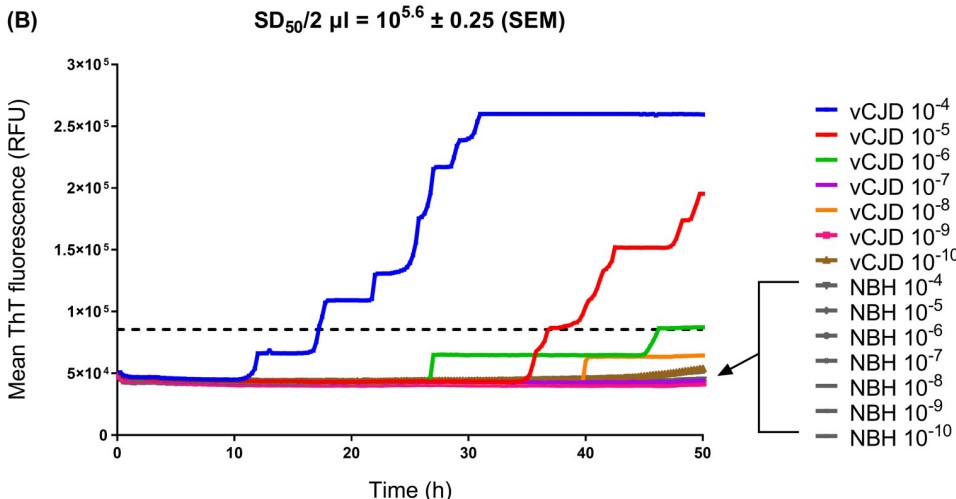

**Fig 1. Endpoint dilution of prion-infected brain homogenates by RT-QuIC.** Representative RT-QuIC experiments testing a ten-fold dilution series of (A) reference BSE-infected sheep brain tissue and brain tissue from mock-infected negative control sheep (NBH), and (B) reference vCJD-infected human brain tissue and negative control human brain tissue. RT-QuIC was performed using truncated ovine recPrP (residues 94–233) as a substrate. Fluorescence measurements were plotted over 50 h. Data points represent the mean ThT fluorescence from n = 10 replicates (for prion-infected brain dilutions), or n = 2 replicates (for negative control brain dilutions). A total of 14 negative control reactions were tested per RT-QuIC plate (as indicated by arrows). $SD_{50}$ values ± standard error of the mean (SEM) shown in boxes. Horizontal dashed lines indicate threshold fluorescence.

indicating that the limit of detection for the RT-QuIC assay was reached at an approximate $10^{-7}$–$10^{-8}$ dilution of BSE-infected brain tissue.

## Comparison of the analytical sensitivity of RT-QuIC and PMCA with mouse bioassay using BSE-infected sheep brain samples

The same reference brain homogenate pool from BSE-positive sheep (BSB/7/10) was used to compare the analytical sensitivity of RT-QuIC with that of an established PMCA assay, termed

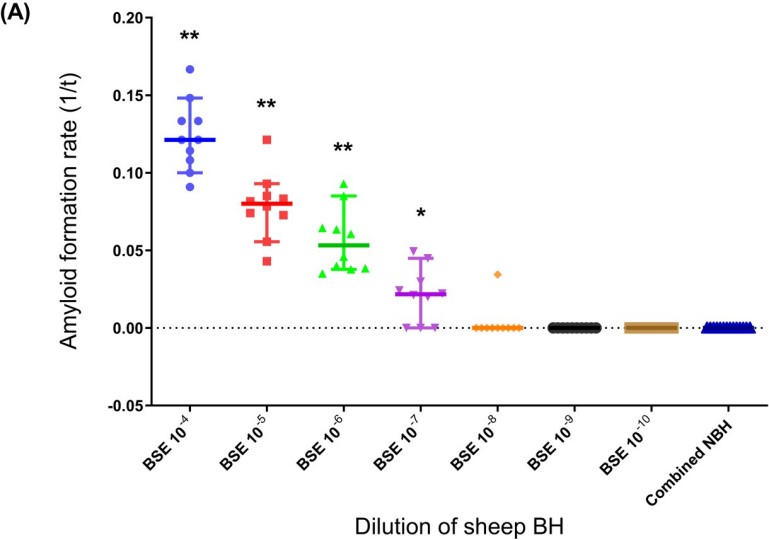

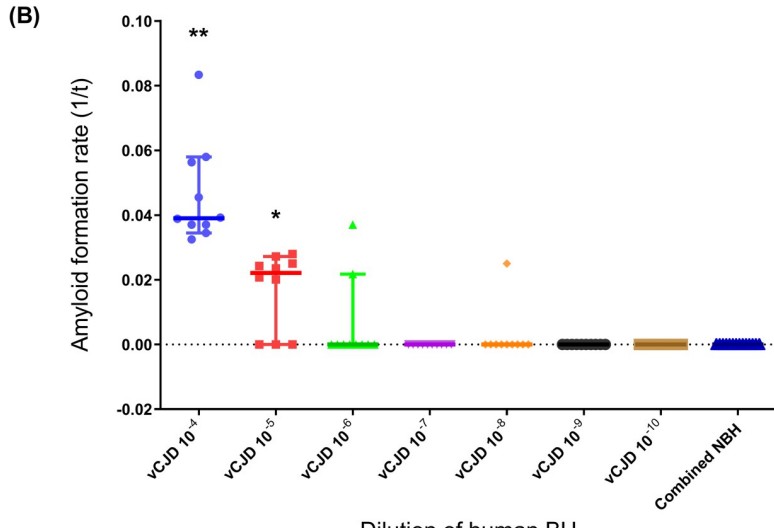

**Fig 2. RT-QuIC prion seeding activity, as determined by endpoint dilution of prion-infected brain homogenates.**
Amyloid formation rates ($1/t$) were plotted for the RT-QuIC experiments shown in Fig 1, testing a ten-fold dilution series of (A) reference BSE-infected sheep brain tissue and brain tissue from mock-infected negative control sheep (NBH), and (B) reference vCJD-infected human brain tissue and negative control human brain tissue. Median rates for each brain dilution (n = 10 replicates/dilution) are shown as solid horizontal lines on scatterplots; solid vertical lines show 95% confidence intervals. In both sets of experiments, negative controls (NBH) were combined (n = 14) as no dilutional effect was apparent. Statistically significant difference in amyloid formation rate between a given dilution of prion-infected brain *vs*. negative controls determined by Wilcoxon signed-rank test, indicated as follows: * ($p \leq 0.05$); ** ($p \leq 0.01$).

"miniaturized bead-PMCA" (mb-PMCA), which was previously shown to detect PrP$^{Sc}$ in BSE-infected sheep blood samples [32,50]. The limit of detection for each assay was compared by endpoint dilution of the reference brain homogenate (Table 1). SD$_{50}$ values (seeding dose $_{50}$ = dilution of brain homogenate at which 50% replicates test positive in the assay) were

calculated using the Spearman-Kärber method, Eqs (1) & (2), to provide a quantitative comparison of the relative sensitivity of each assay [36].

Three independent experiments were run with RT-QuIC and mb-PMCA, each including 10 replicate reactions per brain dilution (S1 and S2 Tables). The mean $\log_{10}$ $SD_{50}$ value calculated from mb-PMCA was 9.43 ± 0.05 (± standard deviation, StDev), equating to $10^{11.7}$ $SD_{50}$ units/g brain tissue, with the limit of detection (endpoint dilution) being reached at a brain dilution of $10^{-10}$ (equivalent to approximately 500 fg brain tissue) (Table 1). For RT-QuIC, the mean $\log_{10}$ $SD_{50}$ value was 7.37 ± 0.05 (± StDev) equivalent to $10^{10.1}$ $SD_{50}$ units/g of pooled brain tissue, with the limit of detection being reached at a brain dilution of $10^{-8}$ (equivalent to approximately 20 pg brain tissue) (Table 1).

Bioassay remains the gold standard for detection and quantification of prion infectivity in biological samples [61]. In bioassays, the amount of infectious agent is estimated by inoculation of groups of mice/rodents with serial dilutions of brain homogenate, and calculation of the $ID_{50}$ (infectious dose $_{50}$ = dilution of brain homogenate at which 50% of inoculated animals become infected). $SD_{50}$ values, calculated by RT-QuIC and mb-PMCA, were compared with the $ID_{50}$ from bioassay of the same reference brain homogenate from BSE-infected sheep (BSB/7/10) in a transgenic mouse line expressing approximately wild type levels of sheep PrP (*PRNP* allele ARQ) (tgOvARQ mice) [51]. This mouse line was used because it was predicted to be sensitive to infection with sheep-adapted BSE, and offers an alternative to over-expressing transgenic mouse lines, in which high levels of PrP expression have been linked to the development of spontaneous neurological disease [62–64]. The $\log_{10}$ $ID_{50}$ value calculated from tgOvARQ endpoint dilution bioassay was 4.7 ± 0.21 (± standard error, SE) (S3 Table), equivalent to approximately $10^{6.4}$ $ID_{50}$ units/g pooled brain tissue, with an endpoint dilution of $10^{-5}$ (equivalent to approximately 200 ng brain tissue) (Table 1). Taken together, these results indicate that the analytical sensitivity of mb-PMCA is approximately 40-fold greater than that of RT-QuIC (S1 and S2 Tables), whereas both *in vitro* assays are more sensitive than the tgOvARQ bioassay, in which the limit of detection was 3 and 5 $\log_{10}$-fold lower than that of RT-QuIC or mb-PMCA, respectively (Table 1).

## Comparison of analytical sensitivity of RT-QuIC and PMCA on vCJD-infected human brain samples

The mb-PMCA and RT-QuIC assays described in this study were both optimised for detection of PrP$^{Sc}$ in BSE-infected sheep tissues, but if they are to be applied in human medicine, it is also important to consider their ability to detect PrP$^{Sc}$ in samples from patients with prion diseases. We therefore ran additional endpoint dilution experiments for each method, using a reference vCJD-infected human brain homogenate (NHBY0/0003) as seed (S1 and S2 Tables).

For mb-PMCA, the mean $\log_{10}$ $SD_{50}$ value was 7.85 (n = 2 experiments) (S2 Table), equivalent to approximately $10^{10.2}$ $SD_{50}$ units/g brain tissue, with a limit of detection being reached at a brain dilution of $10^{-9}$ (Table 1), equivalent to approximately 5 pg brain tissue. For RT-QuIC, the mean $\log_{10}$ $SD_{50}$ value was 5.43 ± 0.24 (± StDev, n = 3) (S1 Table), equivalent to approximately $10^{8.1}$ $SD_{50}$ units/g brain tissue. The limit of detection for RT-QuIC was reached at an approximate $10^{-5}$–$10^{-6}$ dilution of vCJD brain (Figs 1B and 2B and Table 1), equivalent to approximately 2–20 ng brain tissue.

These results show that the analytical sensitivity of mb-PMCA is higher than that of RT-QuIC when applied to vCJD-infected human brain tissue, which is consistent with the results of experiments using the reference BSE-infected sheep brain homogenate (Table 1). These data also suggest that the sensitivity of detection of PrP$^{Sc}$ is marginally lower in human vCJD-infected brain tissue compared to sheep BSE-infected brain tissue for both mb-PMCA

**Table 1. Comparison of the limit of detection for *in vitro* assays and mouse bioassay.**

| Assay | Seed | Dilution of prion-infected brain homogenate | | | | | | | | | | |
|---|---|---|---|---|---|---|---|---|---|---|---|---|
| | | $10^{-1}$ | $10^{-2}$ | $10^{-3}$ | $10^{-4}$ | $10^{-5}$ | $10^{-6}$ | $10^{-7}$ | $10^{-8}$ | $10^{-9}$ | $10^{-10}$ | $10^{-11}$ |
| RT-QuIC | vCJD | ND | ND | ND | 1 | 0.7 | 0.1 | 0 | 0 | 0 | 0.1 | ND |
| | ShBSE | ND | ND | ND | 1 | 1 | 1 | 0.7 | 0.1 | 0 | 0 | ND |
| mb-PMCA | vCJD | ND | ND | ND | 1 | 1 | 1 | 1 | 0.3 | 0.1 | 0 | 0 |
| | ShBSE | ND | ND | ND | ND | 1 | 1 | 1 | 1 | 0.8 | 0.2 | 0 |
| Bioassay (tgOvARQ) | ShBSE | 0.9 | 1 | 1 | 0.9 | 0.3 | 0 | 0 | 0 | ND | ND | ND |

Data represent mean proportion of positive replicates at each brain dilution, as tested by RT-QuIC (n = 3 independent experiments) and mb-PMCA (n = 2 or 3 independent experiments for vCJD and ShBSE datasets, respectively), or proportion of infected mice at each dilution (for tgOvARQ bioassay) (see S1–S3 Tables). Dilutions with positive replicates/mice are shaded grey. Endpoint dilutions, shaded dark grey, indicate the approximate limit of detection for each assay; ND, not determined.

and RT-QuIC (Table 1 and S1 and S2 Tables). However, it is important to note that due to species-specific and/or prion strain-specific differences in brain infectivity levels, the starting concentration of $PrP^{Sc}$ in these two brain samples may not have been directly comparable.

## Optimisation of RT-QuIC to detect $PrP^{Sc}$ in BSE-infected sheep blood samples

Several RT-QuIC amplification strategies have been used to assess blood [42,43,46,48], and spiked blood fractions [40] for prion-associated amyloid seeding activity. However, to the best of our knowledge, there is no published data on the application of RT-QuIC to endogenously-infected blood samples from vCJD patients, or its application to blood samples from a large animal model of vCJD. Therefore, we have extended on previous studies using IOME to enhance sensitivity of prion detection by RT-QuIC [45,46], to adapt and optimise RT-QuIC for detection of seeding activity in whole blood samples from an established large animal model of vCJD, sheep experimentally infected with BSE. The rationale for optimising our version of RT-QuIC on whole blood samples was that the assay would be more adaptable to high throughput screening if there was no requirement for processing blood into fractions prior to testing.

**Detection of seeding activity in "spiked" blood.** We used the RT-QuIC method described above to test a small panel of whole blood samples from BSE-infected sheep, but failed to obtain any positive results (Fig 3A). Therefore, to increase the sensitivity of detection and establish whether whole blood contained inhibitors of the RT-QuIC reaction, an additional iron oxide magnetic extraction (IOME) step with superparamagnetic iron oxide beads (IOBs) was added during sample preparation, based on a methodology developed by Denkers *et al.*, in which IOBs were shown to capture and concentrate amyloid seeding activity in complex biological samples, due to the avid metal binding property of prions [45]. Samples of spiked blood (i.e. a spike of BSE-infected sheep brain homogenate that had been serially diluted into uninfected whole blood) were incubated in the presence of IOBs in a large excess of buffer, to remove potential inhibitors, followed by magnetic separation and washing of IOBs prior to amplification and detection by RT-QuIC.

Using this method, the adapted RT-QuIC assay detected seeding activity within 40 h in blood spiked with $10^{-2}$ to $10^{-5}$ dilutions of BSE-infected sheep brain tissue (Fig 3B) with no signal detected in negative controls i.e. reactions seeded $10^{-2}$ or $10^{-3}$ dilutions of blood spiked with uninfected brain homogenate (please note that the apparently lower sensitivity of detection compared to previous endpoint dilution experiments, exemplified in Fig 1A, may be due to use of a different BSE-infected sheep brain homogenate derived from a single sheep)

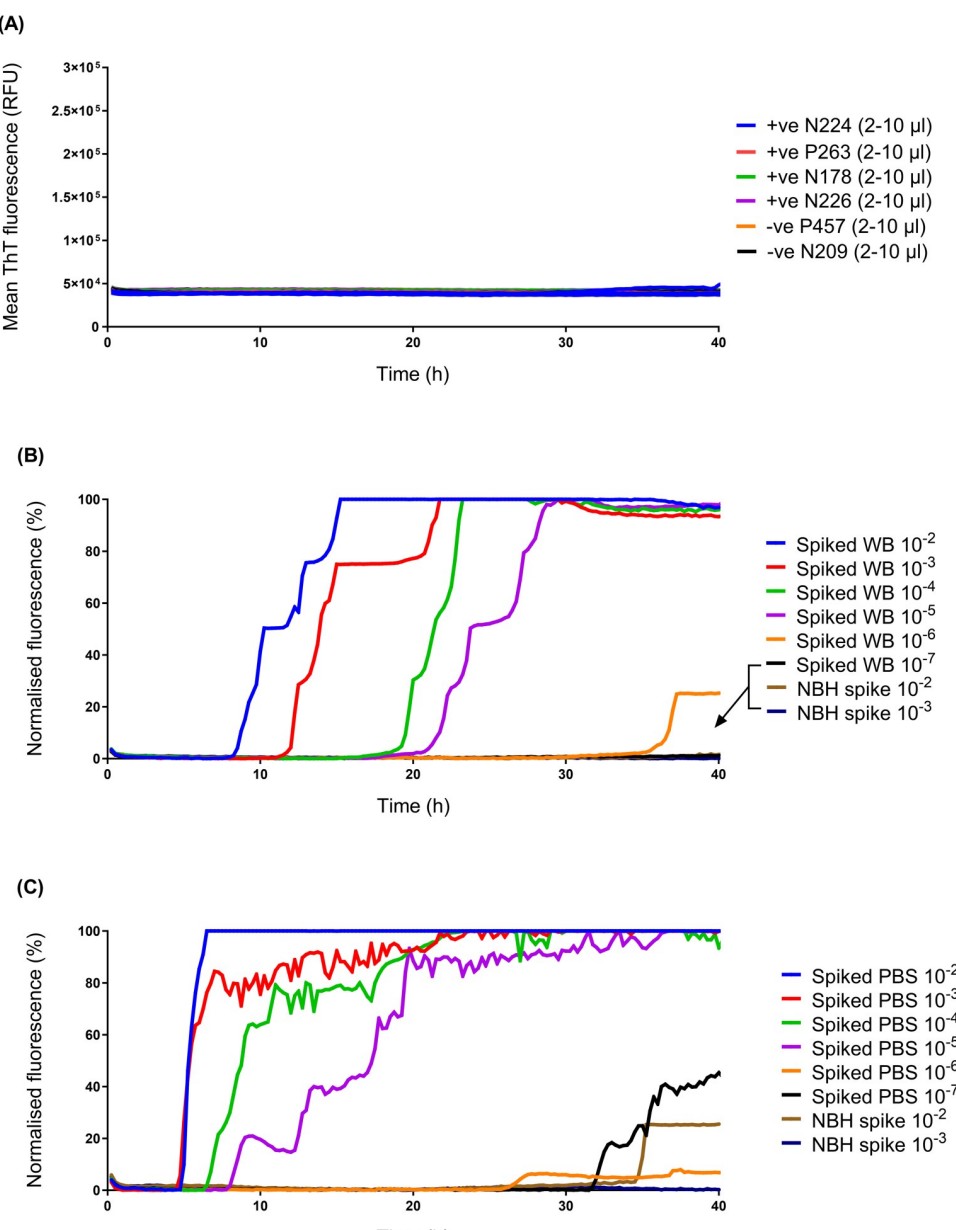

**Fig 3. Adapted RT-QuIC assay including IOME for detection of PrP$^{Sc}$ in spiked whole blood.** (A) Whole blood samples (2 µl, 5 µl or 10 µl) collected from clinical stage, BSE-positive (+ve) sheep (ID numbers: N224, P263, N178, N226) were tested by RT-QuIC, alongside negative control (-ve) samples from mock-infected sheep (ID numbers: P457, N209) but did not yield positive results. (B) Samples (5 µl) of spiked whole blood (Spiked WB; BSE-infected sheep brain homogenate that had been serially diluted into uninfected whole blood) and corresponding negative controls (NBH spike; reactions seeded $10^{-2}$ or $10^{-3}$ dilutions of blood spiked with uninfected brain homogenate) were tested using an adapted RT-QuIC assay, in which spiked blood samples underwent IOME. (C) A dilution series of PBS spiked with the same BSE-infected and uninfected brain homogenates (labelled "spiked PBS" and "NBH spike", respectively) was run alongside for comparison. Data were normalised to allow comparison between datasets with heterogeneous baseline fluorescence. Normalised fluorescence, $F-F_{min}/F_{max}-F_{min}$: F, the mean fluorescence signal from n = 4 replicates; $F_{min}$, the smallest fluorescence value for each dataset; $F_{max}$, maximum (saturating) fluorescence (equivalent to 260,000 RFU). Data were expressed as a percentage, with $F_{min}$ set as 0% and $F_{max}$ set at 100%.

(Fig 3B). The adapted assay showed a comparable analytical sensitivity when seeded with samples of the same sheep brain homogenate diluted in PBS (seeding activity detected in $10^{-2}$ to $10^{-5}$ dilutions of spiked PBS within 40 h) (Fig 3C), indicating that IOME pre-treatment had minimised any inhibitory effects of whole blood components on the RT-QuIC reaction.

**Detection of seeding activity in blood samples containing endogenous BSE infectivity.** Initial experiments to determine whether the IOME adapted RT-QuIC assay could also detect seeding activity in endogenously infected blood samples (i.e. samples collected from sheep experimentally infected with BSE), did not produce positive results (S4 Fig), suggesting that the endogenously infected blood samples contained lower levels of PrP$^{Sc}$ (compared to spiked blood samples). To further increase the sensitivity of the RT-QuIC assay, a larger starting volume (200 μl) of blood was diluted in an optimised "IOME capture buffer" containing detergent, nuclease, protease inhibitors and bovine serum albumin (BSA) as a carrier protein; a modified version of the capture buffer developed for a different blood test for vCJD, the "direct detection assay" (DDA) [65]. Sonication steps were also added to the IOME pre-treatment protocol to reduce aggregation of blood components on IOBs, which tended to occur when larger volumes of blood were used.

Following these optimisation steps, the modified RT-QuIC assay, hereafter referred to as "whole blood IOME RT-QuIC" (WB IOME RT-QuIC) was again tested on blood samples from BSE-infected and mock-infected sheep to determine analytical parameters for the assay. A cut-off time of 20 h was chosen, based on the observation that endogenously-infected whole blood samples from positive control animals tended to amplify within this timeframe (Fig 4A), as did serial dilutions of individual blood samples (Fig 4B), whereas the rate of unseeded fibrilisation increased substantially when the reaction was run for longer (Fig 4C). Although we rarely observed spontaneous ThT-positive fibrilisation in negative control reactions before 20 h (Fig 4C), a stringent threshold value of 75% maximum (saturating) ThT fluorescence (75% F$_{max}$, equivalent to 195,000 RFU) was chosen to minimise potential false positives, as determined by receiver operator characteristic (ROC) analysis (S5 Fig). Thus, blood samples were scored positive if at least 50% replicate reactions exceeded threshold fluorescence within the first 20 hours of the experiment.

As proof of principle, WB IOME RT-QuIC was used to test a small panel of whole blood samples from sheep that had been infected experimentally with BSE either *via* the oral route (donors) or *via* the intravenous route by blood transfusion (recipients) (Fig 5). The panel included samples collected at different time points ranging from approximately 24 months before disease onset to the clinical stages of infection (representing 37.5% - 100% of survival period), as well as samples from mock-infected negative controls (Table 2). WB IOME RT-QuIC gave positive results in nine out of twelve whole blood samples from BSE-infected sheep (Fig 5 and Table 2). Eleven equivalent buffy coat samples (collected from the same animals at the same time points) previously tested positive by mb-PMCA [32], whereas only eight of the eleven corresponding whole blood samples tested positive by WB IOME RT-QuIC (Table 2). This suggests that mb-PMCA is marginally more sensitive than WB IOME RT-QuIC in detecting PrP$^{Sc}$ in endogenously infected blood, consistent with the greater analytical sensitivity of mb-PMCA (as reported in Table 1).

Interestingly, samples collected from three sheep (N178, N224, P263) at the clinical phase of infection, had substantially longer lag times (and slower amyloid formation rates) in WB IOME RT-QuIC than samples collected from the same animals at an earlier time point. For two animals (N178, N224), differences in lag time between clinical and 15 or 18 mpi samples were statistically significant (p ≤ 0.01, Mann-Whitney unpaired t-test) (Table 2). If we assume that lag time is proportional to PrP$^{Sc}$ concentration, these results are consistent with a decline in the concentration of PrP$^{Sc}$ in blood during the clinical stages of infection.

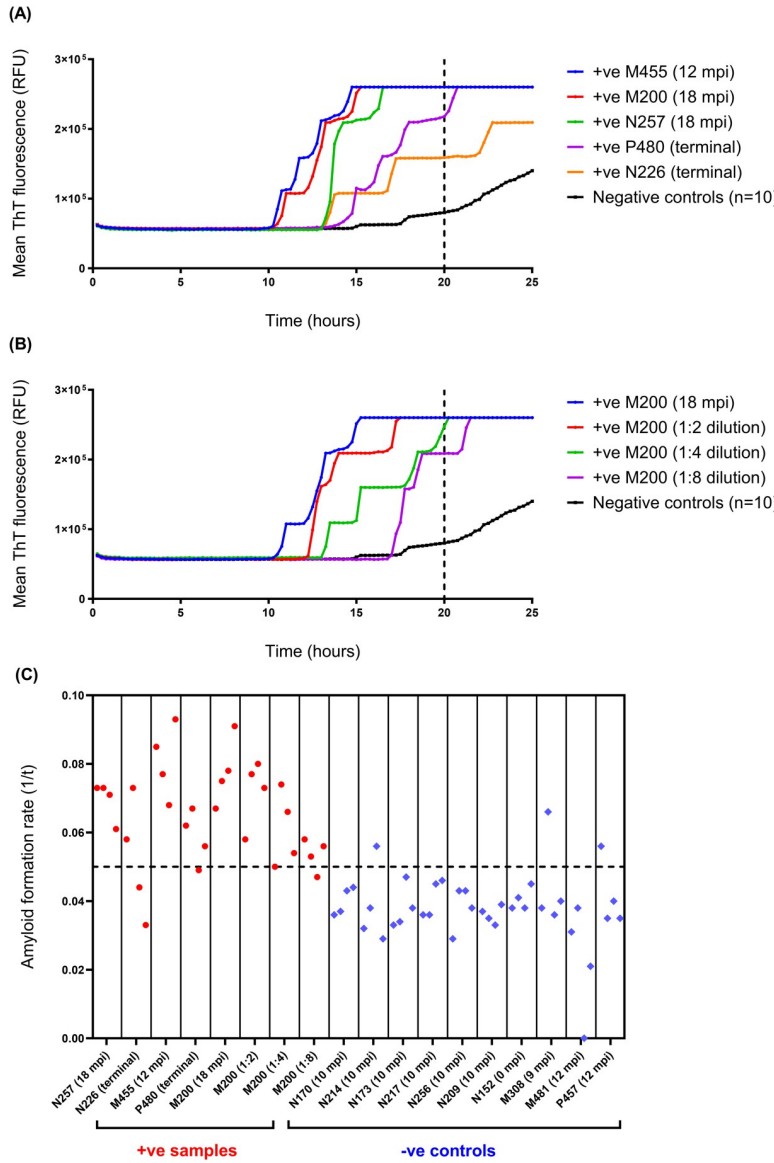

**Fig 4. Optimisation of WB IOME RT-QuIC to determine cut-off time.** (A) Samples of whole blood from presymptomatic sheep (12–18 months post inoculation, mpi) that went on to develop clinical signs, and samples from symptomatic/clinical stage (terminal) BSE-infected (+ve) sheep typically tested positive by WB IOME RT-QuIC within 20 h, (B) including a dilution series of BSE-infected blood from one sheep (ID number: M200). The mean fluorescence from n = 10 negative controls (whole blood samples from mock-infected sheep) is plotted in black. Most negative controls did not display evidence of spontaneous fibrilisation until after 20 h (as indicated by vertical dotted lines). Each blood sample was tested in n = 4 replicate reactions. (C) Amyloid formation rates ($1/t$) were plotted for the aforementioned BSE-infected (+ve) and negative (-ve) control samples, over a period of 50 h. Rates for individual replicate reactions are plotted, including n = 32 prion positive reactions (plotted in red) and n = 40 negative control reactions (plotted in blue). Cut-off time was set at 20 h (equivalent to $1/t = 0.05$, as indicated by horizontal dotted line) as positive replicates tended to amplify within this time frame ($1/t \geq 0.05$), whereas most negative control reactions did not show evidence of spontaneous fibrilisation until after this time ($1/t < 0.05$). Although a small number of negative control reactions amplified before 20 h (3/40 reactions, in this example), this was only observed in 1/4 replicate reactions for a given blood sample. Thus, a positive sample was defined as one in which $\geq$ 50% (2/4) replicate reactions exceeded threshold fluorescence within a 20 h cut-off time.

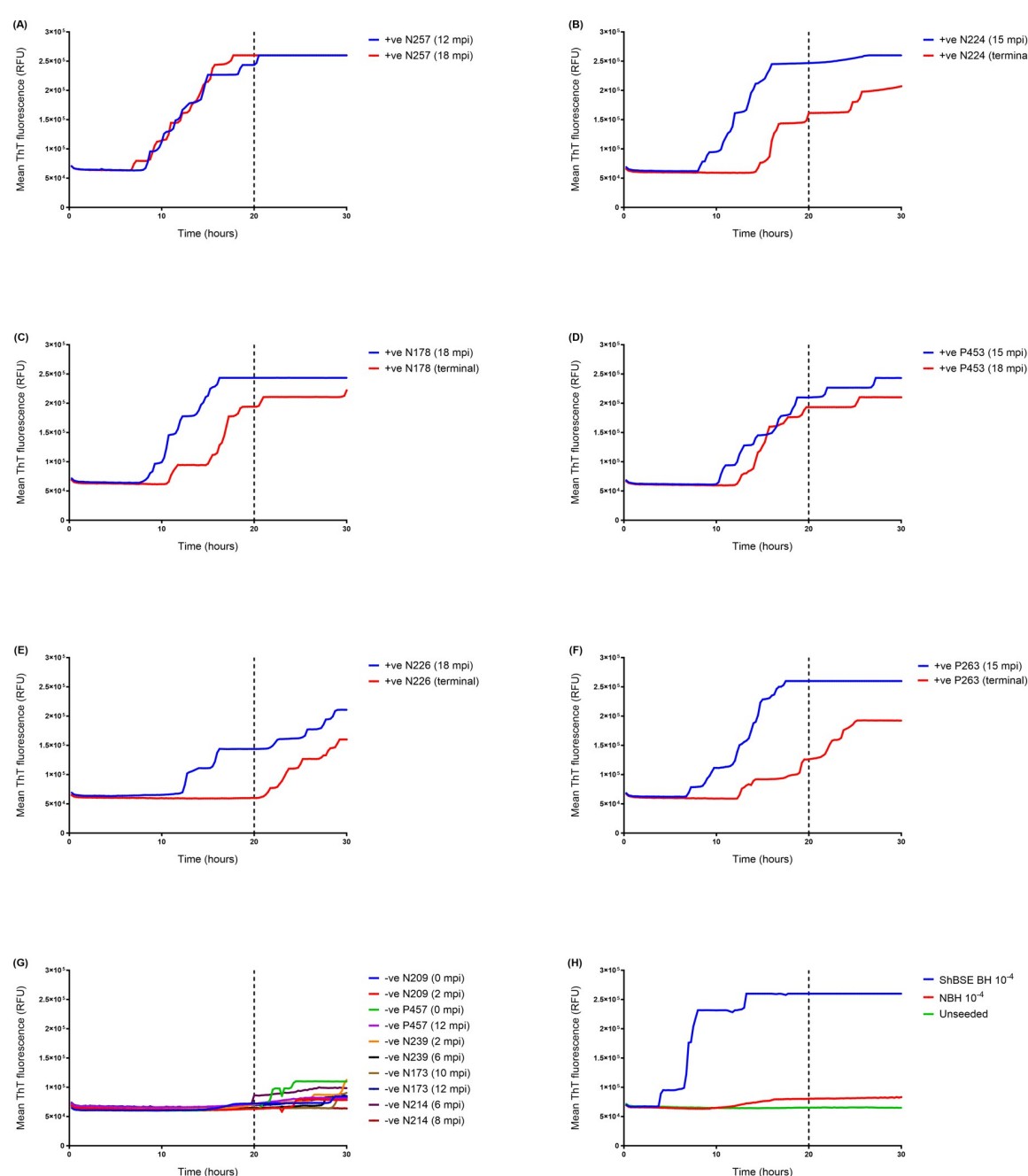

**Fig 5. WB IOME RT-QuIC analysis of a panel of blood samples from BSE-infected sheep (endogenous infectivity).** Samples of whole blood (n = 12) from BSE positive (+ve) sheep collected at different time points (as indicated on figure) were tested by WB IOME RT-QuIC in three independent experiments (equivalent to a total of n = 12 replicate reactions/sample). Blood samples from BSE-infected donor sheep (A) N257, (C) N178, and (E) N226 were tested alongside samples from BSE-infected recipients (B) N224, (D) P453, and (F) P263. Samples were considered positive if ≥ 50% (6/12) replicate reactions exceeded threshold fluorescence within a designated cut-off time of 20 h (as indicated by vertical dotted lines). (G) Whole blood samples (n = 10) from mock-infected, negative (-ve) control sheep were run alongside. A total of n = 128 negative control reactions were tested; samples N209 (2 mpi), P457 (12 mpi), N239 (6 mpi), N173 (12 mpi), N214 (6 mpi) and N214 (8 mpi) were tested in three independent experiments (equivalent to n = 12 replicate reactions/ sample); samples N209 (0 mpi), P457 (0 mpi) and N239 (2 mpi) were tested in four independent experiments (n = 16 replicate reactions/ sample); and sample N173 (10 mpi) was tested in two independent experiments (n = 8 replicate reactions).

**Table 2. Results of WB IOME RT-QuIC testing of a panel of whole blood samples from BSE-infected sheep.**

| Sheep ID (infection route) [a] | Sample time point (% survival period) [b] | Sample status | PMCA result [c] | WB IOME RT-QuIC result | WB IOME RT-QuIC Number (%) positive replicates [d] | Mean lag time (*t*) hours (± StDev) |
|---|---|---|---|---|---|---|
| N239 (D) | 2 mpi (3%) | NC | ND | - | 0/16 (0%) | NA |
|  | 6 mpi (9%) | NC | ND | - | 0/12 (0%) | NA |
| N209 (D) | 0 mpi (0%) | NC | ND | - | 1/16 (6%) | 17 |
|  | 2 mpi (3%) | NC | ND | - | 0/12 (0%) | NA |
| N173 (D) | 10 mpi (13%) | NC | ND | - | 0/8 (0%) | NA |
|  | 12 mpi (15%) | NC | ND | - | 0/12 (0%) | NA |
| N214 (D) | 6 mpi (11%) | NC | ND | - | 1/12 (8%) | 20 |
|  | 8 mpi (15%) | NC | ND | - | 0/12 (0%) | NA |
| N257 (D) | 12 mpi (59%) | BSE | + | + | 11/12 (92%) | 12.5 (± 3.1) |
|  | 18 mpi (88%) | BSE | + | + | 12/12 (100%) | 12.6 (± 3.1) |
| N178 (D)** | 18 mpi (47%) | BSE | + | + | 11/12 (92%) | 12.2 (± 2.5) |
|  | Clinical (100%) | BSE | + | + | 8/12 (67%) | 15.5 (± 2.8) |
| N226 (D) | 18 mpi (63%) | BSE | + | - | 5/12 (42%) | 14.2 (± 1.8) |
|  | Clinical (100%) | BSE | + | - | 0/12 (0%) | NA |
| P457 (R) | 0 mpi (0%) | NC | ND | - | 0/16 (0%) | NA |
|  | 12 mpi (15%) | NC | ND | - | 0/12 (0%) | NA |
| N224 (R)** | 15 mpi (72%) | BSE | + | + | 11/12 (92%) | 12.4 (± 2.4) |
|  | Clinical (100%) | BSE | + | + | 6/12 (50%) | 16.6 (± 1.8) |
| P453 (R) | 15 mpi (38%) | BSE | ND | + | 9/12 (75%) | 14.6 (± 3.1) |
|  | 18 mpi (45%) | BSE | + | + | 8/12 (67%) | 15.6 (± 2.1) |
| P263 (R) | 15 mpi (75%) | BSE | + | + | 12/12 (100%) | 13.1 (± 3.1) |
|  | Clinical (100%) | BSE | + | - | 4/12 (33%) | 16.3 (± 3.4) |

NC, negative control; BSE, BSE-positive; ND, not determined; +, positive for seeding activity; -, negative for seeding activity; StDev, standard deviation; NA, not applicable; mpi, months post inoculation. Blood samples were tested in three independent experiments (giving a total of n = 12 replicate reactions/sample), apart from negative controls, N209 (0 mpi), P457 (0 mpi) and N239 (2 mpi), which were tested in four independent experiments (n = 16 replicate reactions/sample), and N173 (10 mpi), which was tested in two independent experiments (n = 8 replicate reactions).

[a] Infection route: Donors (D) *via* oral route, recipients (R) *via* blood transfusion.

[b] % survival period = time from infection to blood collection x 100/time from infection to culling.

[c] Previously published by Salamat *et al.* (2021) [32].

[d] Number of positive replicates displayed as the number of positive replicates/total number of replicates tested for that sample.

** Sheep in which the clinical stage sample had a significantly longer lag time compared to sample from earlier time point (p ≤ 0.01, Mann-Whitney unpaired t-test).

## Discussion

Although the number of vCJD cases has been in decline since the early 2000s [15], uncertainties surrounding the number of asymptomatic (preclinical or subclinical) "carriers" of vCJD and their potential to infect others, means that secondary (human-to-human) transmission of the disease remains a concern for public health. A number of costly risk reduction measures were implemented in the UK to safeguard the blood supply, including leucoreduction of all labile blood products, and restrictions on blood donations (donor deferral) [66]. The necessity for many of these precautionary measures would be removed if there was a highly sensitive, validated, high-throughput blood test for vCJD that could reliably identify asymptomatic carriers of infection and be applied to screen donated blood.

The aim of this study was to develop a prototype diagnostic/screening test, based on the real-time quaking-induced conversion (RT-QuIC) assay, which would be sensitive enough to

detect preclinical prion infection in blood samples. As preclinical blood samples from vCJD patients are extremely rare, the assay was developed and optimised using blood samples from an established large animal model of vCJD, sheep experimentally infected with BSE [32,49,67,68]. The BSE-infected sheep model, which was previously used to study the risks of transmitting prion disease by blood transfusion [32], provides certain advantages over other animal models (e.g. mouse models); blood can be collected and processed into clinically relevant blood components, in similar volumes to those used in human blood transfusion. In addition, the distribution of PrP$^{Sc}$ in lymphoid tissue and peripheral pathogenesis of BSE-infected sheep closely resembles that of vCJD patients [6,52].

Our version of the RT-QuIC assay, which used truncated sheep recPrP as substrate, showed a high level of analytical sensitivity when tested on a reference BSE-infected sheep brain homogenate, with an estimated SD$_{50}$ of $10^{10.1}$ SD$_{50}$ units/g brain. When compared to the results of endpoint titration experiments of the same reference brain homogenate by mb-PMCA and intracerebral inoculation in tgOvARQ mice, RT-QuIC had a lower analytical sensitivity than mb-PMCA ($10^{11.7}$ SD$_{50}$ units/g) but higher sensitivity than the mouse bioassay ($10^{6.4}$ ID$_{50}$ units/g). These findings are in agreement with other studies showing greater analytical sensitivity of PMCA and RT-QuIC in comparison to bioassays [69]. A possible explanation is that infectious PrP$^{Sc}$ may comprise only a subset of the total misfolded PrP species responsible for seeding aggregation in PMCA and RT-QuIC. Fractionation of PrP$^{Sc}$ extracted from scrapie-infected hamster brain showed that the highest levels of infectivity were associated with particles comprising 14–28 PrP molecules, but did not specifically consider which fractions promoted conversion in RT-QuIC or PMCA [70]. A recent detailed study comparing infectivity, RT-QuIC seeding activity and PrP$^{Sc}$ levels in BSE-infected mouse brains concluded that seeding activity correlated more closely with PrP$^{Sc}$ levels than infectivity [71]. However, further work is required to precisely define which PrP$^{Sc}$ species are involved in seeding PrP conversion in RT-QuIC and PMCA assays, respectively.

The transgenic mouse line (tgOvARQ) used in bioassay experiments was derived from the Tg(OvPrP-A136)3553 line, which when crossed with PrP-null mouse lines produces hemizygous offspring that express sheep PrP$^{C}$ under the control of the mouse *Prnp* promoter, at similar levels to those in the central nervous system of wild-type mice [51]. This line may offer certain advantages to alternative transgenic mouse models that over-express PrP, since they more accurately recapitulate normal PrP$^{C}$ expression in neurons [51], and are less likely to develop spontaneous neurological dysfunction, as reported for some over-expressing transgenic mouse lines [62–64]. The reference BSE-infected sheep brain homogenate used in this study has not been titrated in other mouse lines, and therefore we cannot make a direct comparison of the sensitivity of the tgOvARQ bioassay with more established models. However, comparison with published data on infectivity titres in BSE-infected sheep brains suggests that bioassay in tgOvARQ mice (endpoint dilution of $10^{-5}$) may be marginally less sensitive than in TgBov/tg110 mice (transgenic mice overexpressing bovine PrP$^{C}$) or TgShpXI mice (transgenic mice overexpressing sheep ARQ PrP$^{C}$), with endpoint dilutions of $10^{-6}$ and $10^{-7}$, respectively [32,72], but more sensitive than in wild-type RIII mice (endpoint dilution of $10^{-4}$) [73].

In previous studies, RT-QuIC has been shown to detect low levels of PrP$^{Sc}$ in blood from preclinically infected hamsters [40,42,43] and cervids [42,43,46], and in plasma from scrapie-infected mice [48], using a variety of pre-amplification strategies to concentrate or capture PrP$^{Sc}$ in order to enhance detection. These include sodium phosphotungstate (NaPTA) precipitation [42,43], lipase treatment [46], immunoprecipitation [40,48], or IOME [46]. Modifications to further enhance sensitivity of detection by RT-QuIC include addition of a substrate replacement step to the RT-QuIC reaction [40,44], or using sequential PMCA and RT-QuIC amplification steps [46,74].

To adapt the RT-QuIC method for detection of PrP$^{Sc}$ in small volumes (0.2 ml) of whole BSE-infected sheep blood, an optimised iron oxide magnetic extraction (IOME) pre-treatment step was incorporated into sample preparation [45]. The resulting assay, termed "whole blood IOME RT-QuIC" (WB IOME RT-QuIC), was able to accurately identify blood samples collected from BSE-infected sheep at both clinical and preclinical time points up to 2 years before disease onset. Interestingly, some samples from the clinical stage of disease showed lower amplification efficiencies than samples from the same sheep at preclinical stages of infection, suggesting that PrP$^{Sc}$ levels in blood may decline after the onset of clinical signs.

The preliminary data from a small panel of BSE-infected sheep blood samples presented in this study suggest that the WB IOME RT-QuIC assay has marginally lower diagnostic sensitivity than mb-PMCA, consistent with the lower analytical sensitivity of RT-QuIC demonstrated in comparative titrations of prion-infected brain homogenate. The next step should be a more extensive comparison of diagnostic sensitivity and specificity of WB IOME RT-QuIC compared to mb-PMCA and other blood-based assays for vCJD, testing a larger, blinded panel of BSE-infected and uninfected sheep blood samples. Other diagnostic assays tested may include "plasminogen-bead capture PMCA", which was previously shown to detect blood samples from vCJD patients with high levels of sensitivity and specificity, including a small number of preclinical samples [34], and the "direct detection assay" (DDA), an ELISA-style immunoassay that captures PrP$^{Sc}$ on stainless steel particles, which has also been shown to detect endogenous vCJD in patient blood samples [65].

It would also be interesting to see how the sensitivity of WB IOME RT-QuIC compares to other RT-QuIC blood assays, such as "enhanced RT-QuIC" (eQuIC), in which seeding activity in blood (plasma, serum) is captured by immunoprecipitation on 15B3-coated beads, prior to RT-QuIC with hamster-sheep chimeric recPrP and a substrate replacement step. The eQuIC assay was previously shown to enhance detection of human vCJD brain tissue spiked into human plasma to an endpoint dilution of $10^{-14}$ [40], and was also used to detect endogenous prionemia in clinical and preclinical blood samples from scrapie-infected rodents [40,48]. Furthermore, it would be interesting to see whether sensitivity of the WB IOME RT-QuIC assay can be improved by inclusion of recPrP substrates that were not tested in the current study, such as bank vole recPrP, which has been shown to act as an apparently universal substrate for RT-QuIC [75]. Further gains in sensitivity may also be possible by adapting WB IOME RT-QuIC for use on blood components, such as buffy coat fraction.

RT-QuIC has certain technical advantages over PMCA that may make it more amenable to high-throughput diagnostic or screening applications. In particular, the use of recombinant PrP rather than uninfected mouse brain homogenate as substrate offers huge advantages in terms of standardization, quality control, cost and reduction in experimental animal use. Furthermore, as RT-QuIC measures PrP aggregation in real time through the binding of a fluorescent dye, the assay may be adaptable to faster, high-throughput, quantitative analysis [76].

There have been no published reports to date on the use of RT-QuIC for detection of endogenous seeding activity in blood samples from vCJD patients, or blood samples from a large animal model of vCJD. Our data suggest that WB IOME RT-QuIC and mb-PMCA (both developed for application to BSE-infected sheep samples) could potentially be applied to human samples with minimal modification, as both mb-PMCA and our version of the RT-QuIC reaction show sensitive detection of PrP$^{Sc}$ in a reference vCJD-infected brain homogenate, to endpoint dilutions of $10^{-9}$ and $10^{-6}$, respectively. For each assay, the endpoint dilutions in the vCJD titration experiments were 1–2 $\log_{10}$ lower than in titrations of the reference BSE-infected sheep brain. This may reflect slightly lower analytical sensitivities of RT-QuIC and mb-PMCA due to species differences in PrP sequence, but could also simply be due to differences in PrP$^{Sc}$ concentrations in the two reference brain homogenates. The next

logical step would be to test the ability of WB IOME RT-QuIC to identify infected blood samples from vCJD patients. Due to the paucity of such human blood samples, this will be dependent on satisfying the strict criteria governing release of samples, and the priorities of public health authorities and policymakers.

## Conclusions

The results from this study show that WB IOME RT-QuIC permits sensitive detection of PrP$^{Sc}$ in whole blood from BSE-infected sheep at clinical and preclinical time points, and suggest that the assay may be applicable to tissue/blood samples from vCJD patients. In light of the need for a rapid, non-invasive, high-throughput test to detect asymptomatic carriers of vCJD, these promising results support the further development and evaluation of WB IOME RT-QuIC for screening of donated blood and/or diagnosis.

## Supporting information

**S1 Fig. Optimisation of RT-QuIC to detect PrP$^{Sc}$ in BSE-infected sheep brain tissue.** Different substrates were tested by RT-QuIC, including (A) truncated ovine recPrP (amino acids 94–233. *PNRP* genotype ARQ), (B) full length ovine recPrP (resi 25–233. PNRP genotype ARQ), (C) hamster-sheep recPrP (resi 23–137 Syrian golden hamster PrP followed by resi 141–234 ovine PrP, *PNRP* genotype ARQ), and (D) full length bovine recPrP (resi 25–241). (E-H) Further experiments were performed to determine the optimal concentration of SDS required for RT-QuIC using truncated ovine recPrP. Based on previous literature, SDS concentrations within the 0.025–0.1% (w/v) range were tested. RT-QuIC reactions were seeded with a $10^{-4}$ dilution of BSE-infected sheep brain homogenate (purple) or brain homogenate from mock-infected negative control sheep (red). Unseeded reactions ("mock-seeded" with PBS buffer) are plotted in blue. In most experiments, a $10^{-4}$ dilution of 263K scrapie-infected hamster brain tissue was used as a positive control (green). Data points represent the mean ThT fluorescence from n = 4 replicates.
(TIF)

**S2 Fig. Expression and purification of recPrP.** (A) Truncated ovine recPrP was expressed in, and purified from, *E. coli* Rosetta (DE3) competent cells (Merck). The process was monitored by analysis of fractions on 12% SDS-PAGE gels, with proteins visualised by Coomassie blue: M, molecular mass marker (units in kDa); U, sample from uninduced cells prior to induction with IPTG; I, sample from induced cells; S, sample from supernatant after clarification by centrifugation; $B_1$, sample taken from buffer while bedding resin in column; $B_2$, sample from flow-through while bedding resin; D, sample from denaturing step with guanidine; $R_1$, sample from gradient refolding; $R_2$, sample form isocratic refolding; $E_7$ and $E_8$, eluted fractions containing recPrP, as evidenced by an approximate 17 kDa protein band corresponding to monomeric truncated ovine recPrP; X, sample from resin cleaning step in which any remaining protein was denatured and stripped from the resin. (B) Samples from eluted fractions were assessed for purity by silver staining: $E_1$-$E_9$, samples from gradient elution with imidazole; $F_1$, final dialysed recPrP stock; $F_2$, final recPrP stock passed through 100 kDa MWCO spin filter. (C) Elution of recPrP was monitored by UV. The fraction corresponding to the middle 50% of the elution peak ($A_{280\ nm} > 0.4$ AU) (Fraction E7/E8, green) was pooled to yield 27 mg recPrP. Original, uncropped and minimally-adjusted gel images are provided in another supplementary information file (S1_raw_images).
(TIF)

**S3 Fig. Determination of 50 h cut-off time for RT-QuIC endpoint dilution experiments.**
Ten-fold dilutions of (A) a reference BSE-infected sheep brain tissue homogenate and brain
tissue from mock-infected negative control sheep (NBH), or (B) a reference vCJD-infected
human brain tissue homogenate and negative control human brain tissue. RT-QuIC was per-
formed using truncated ovine recPrP (residues 94–233) as a substrate. Fluorescence measure-
ments were plotted over 90 h. Data points represent the mean ThT fluorescence from n = 10
replicates (for prion-infected brain dilutions), or n = 2 replicates (for negative control brain
dilutions). False positives start to appear in 1/2 or 2/2 replicates at given negative control brain
dilutions after 50 h (as indicated by arrows).
(TIF)

**S4 Fig. Amplification of endogenously infected blood samples by IOME RT-QuIC.** (A)
Whole blood (WB) and (B) buffy coat (BC) samples (5–10 μl) from BSE-infected (+ve) sheep
(animal ID: N257 at clinical time point) and mock-infected (uninfected) sheep (animal ID:
N214 at 0 mpi) were tested by a modified IOME RT-QuIC assay but failed to produce positive
results. The mean fluorescence from n = 4 replicate reactions is plotted over a period of 80 h.
(TIF)

**S5 Fig. Receiver operating analysis (ROC) to determine threshold fluorescence for WB
IOME RT-QuIC.** A receiver operating characteristic (ROC) curve and corresponding area
under the curve (AUC) was plotted for a representative WB IOME RT-QuIC optimisation
experiment testing eight blood samples from known BSE-positive sheep and ten blood samples
from known negative sheep (n = 4 replicate reactions/sample). The theoretical sensitivity and
specificity of the assay at cut-off (20 h) was plotted for a range of putative threshold values
(range 0–100% $F_{max}$). The optimal threshold value was determined to be 75% $F_{max}$, yielding
100% specificity and 75% sensitivity (indicated by red arrow).
(TIF)

**S1 Table. Summary of RT-QuIC endpoint dilution experiments.** BH, brain homogenate;
Proportion of positive replicates defined as: The number of replicate reactions that exceed
threshold by cut-off time (50 h)/total number of replicates (n = 10) at a given BH dilution. [a]
$Log_{10}$ $SD_{50}$ units (± standard error) present in 2 μg brain calculated according to Spearman-
Kärber method, Eqs (1) & (2). [b] The number of $SD_{50}$ units per g of brain.
(DOCX)

**S2 Table. Summary of mb-PMCA endpoint dilution experiments.** BH, brain homogenate;
Proportion of positive replicates defined as: The number of replicate reactions that scored pos-
itive for PK-resistant $PrP^{Sc}$ by western blot/total number of replicates (n = 10) at a given BH
dilution. [a] $Log_{10}$ $SD_{50}$ units (± standard error) present in 5 μg brain calculated according to
Spearman-Kärber method, Eqs (1) & (2). [b] The number of $SD_{50}$ units per g of brain.
(DOCX)

**S3 Table. Summary of tgOvARQ endpoint dilution bioassay.** BH, brain homogenate; dpi,
days post inoculation; NA, not applicable; Proportion of positive mice = the number of mice
that scored positive for prion infection (by western blot and/or IHC)/total number of mice in
that cohort. [a] $Log_{10}$ $ID_{50}$ units (± standard error) calculated according to Spearman-Kärber
method, Eqs (1) & (2). [b] an estimate of the number of $ID_{50}$ units per g of brain.
(DOCX)

**S1 Raw images. Original images for recPrP purification and elution gels.**
(PDF)

## Acknowledgments

The authors thank staff at the Institute for Animal Health, Compton and the Roslin Institute for animal care and technical assistance. We thank Jillian Cooper and Kaetan Ladhani at the National Institute for Biological Standards and Controls (NIBSC), Medicines and Healthcare products Regulatory Agency (MHRA), and Chris-Anne McKenzie at the MRC Edinburgh Brain and Tissue Bank for providing human brain tissue samples. We thank Glenn Telling (CSU, Colorado, USA) for originally producing and providing the Tg(OvPrP-A136)3533 transgenic mice used in the bioassay experiments, as part of a collaborative project with Nora Hunter. We are also grateful to Olivier Andreoletti (INRAE-ENVT, Toulouse, France) for the generous provision of TgShpXI transgenic mouse brains used as PMCA substrate.

## Author Contributions

**Conceptualization:** Charlotte M. Thomas, M. Khalid F. Salamat, E. Fiona Houston.

**Data curation:** Charlotte M. Thomas, M. Khalid F. Salamat, Christopher de Wolf, Sandra McCutcheon, A. Richard Alejo Blanco, E. Fiona Houston.

**Formal analysis:** Charlotte M. Thomas.

**Funding acquisition:** E. Fiona Houston.

**Investigation:** Charlotte M. Thomas, M. Khalid F. Salamat, Christopher de Wolf, A. Richard Alejo Blanco.

**Methodology:** Charlotte M. Thomas, Sandra McCutcheon, Jean C. Manson, E. Fiona Houston.

**Project administration:** Sandra McCutcheon, Jean C. Manson, Nora Hunter, E. Fiona Houston.

**Resources:** Nora Hunter.

**Software:** Christopher de Wolf.

**Supervision:** Sandra McCutcheon, Jean C. Manson, Nora Hunter, E. Fiona Houston.

**Visualization:** Charlotte M. Thomas.

**Writing – original draft:** Charlotte M. Thomas, E. Fiona Houston.

**Writing – review & editing:** Charlotte M. Thomas, M. Khalid F. Salamat, Sandra McCutcheon, Nora Hunter, E. Fiona Houston.

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
