## [Decision Letter · Decision Letter 0]

31 Jul 2023

PONE-D-23-21489Development of a sensitive real-time quaking-induced conversion (RT-QuIC) assay for application in prion-infected blood.PLOS ONE

Dear Dr. Thomas,

Thank you for submitting your manuscript to PLOS ONE. After careful consideration, we feel that it has merit but does not fully meet PLOS ONE’s publication criteria as it currently stands. Therefore, we invite you to submit a revised version of the manuscript that addresses the points raised during the review process.

We look forward to receiving your revised manuscript.

Kind regards,

Byron Caughey

Academic Editor

PLOS ONE

Journal Requirements:

Additional Editor Comments:

Thank you for submitting your interesting manuscript to PLoS One. Three experts have now reviewed it, and their comments are attached. All of the reviewers indicated that your study is interesting and the subject important. However, two reviewers found that your results so far are a bit preliminary, and that it is important for you to add additional analyses of cases and controls to make this a robust study. One reviewer questioned the inclusion of the PMCA results, but I think that the comparison of assay sensitivities provides some useful context. I will leave up to you to decide what you want to do on that point. Overall, I would ask for a major revision to address the comments of the reviewers (point-by-point).  In particular, you should provide additional data to substantiate your findings as requested by Reviewers 2 & 3.

Reviewers' comments:

Reviewer's Responses to Questions

**Comments to the Author**

1. Is the manuscript technically sound, and do the data support the conclusions?

Reviewer #1: Yes

Reviewer #2: Partly

Reviewer #3: Partly

2. Has the statistical analysis been performed appropriately and rigorously? 

Reviewer #1: Yes

Reviewer #2: Yes

Reviewer #3: Yes

3. Have the authors made all data underlying the findings in their manuscript fully available?

Reviewer #1: Yes

Reviewer #2: Yes

Reviewer #3: Yes

4. Is the manuscript presented in an intelligible fashion and written in standard English?

Reviewer #1: Yes

Reviewer #2: Yes

Reviewer #3: Yes

5. Review Comments to the Author

Reviewer #1: This study is of high scientific value, it has been technically well performed and the flow of the experiments is appropriate. The paper is well written. Although in animals BSE strain might be presumably still circulating and a blood test might be applicable, in humans no vCJD cases are reported since 10 years, except those incidentally infected. In addition, the estimate of vCJD cases infected by transfusion was initially high but then not confirmed. In summary, this is a high quality study, carried out by top level prion scientists but with an unfortunately weakened impact to public health by the disappearance of BSE.

Reviewer #2: Thomas et al describe the use of iron oxide beads followed by RT-QuIC to detect ovine BSE prion seeding activity in ovine whole blood (WB RT-QuIC). This manuscript is well written and describes a promising, albeit very preliminary, approach to the detection of prions in blood.

Major concerns

It is this reviewer’s opinion that, although the findings described in this manuscript are promising, they are still very preliminary. This study only includes a very small number of samples, with only 3 negative control endogenous ovine blood samples in Fig. 4 and Table 2, likely only tested once. Additional testing should be done. This study really needs at the very least testing of an equal number of prion positive and negative samples to confirm the findings and make this a publishable unit. This is the reason this reviewer will request “Major reviews”.

Results from testing of additional negative ovine whole blood samples are reported in Supplementary Figure 4, if these are different animals the data should probably be included in Figure 4.

Minor concerns

This reviewer is a little confused by the inclusion of the PMCA data in this study. The results show that the PMCA is more sensitive than the RT-QuIC but the authors choose to test the endogenous samples with the latter assay, presumably because the RT-QuIC is better suited for practical diagnostic applications. Although the PMCA data is interesting, because it was not used to test endogenous whole blood samples, it just seems to be a confusing addition to the manuscript.

Furthermore, several of the supplementary figures and tables, which contain key information about the development of the WB RT-QuIC and the in vivo data, could be moved into the main text.

Which negatives were included in Fig. 4 ? Please include samples IDs.

The PBS spiking experiments should include negative control brain homogenate spiked PBS.

Reviewer #3: Development of a sensitive real-time quaking-induced conversion (RT-QuIC) assay for application in prion-infected blood.

First and Corresponding Author: Charlotte M. Thomas

Senior Author: Dr. E. Fiona Houston

The authors modified prion amplification assays to develop a hematogenous prion detection assay using a vCJD large animal model (BSE-infected sheep/blood transfusion passage). They provide comparison of modified PMCA and RT-QuIC assays. They demonstrate the utility of mb-PMCA in the detection of blood-borne prions and correlate this to mouse bioassay. Great effort was put forth in recombinant protein optimization, time to positive cutoff, and assay readouts. These components of the work provide great insight into the use of specific recombinant proteins in the detection of prions present in blood. Further exploration of sample pretreatment is warranted as challenges have been encountered in the development of assays that are able to detect prions during the very early phases of disease, especially detection in blood.

The manuscript is exceptionally well written and organized.

Major flaw:

The number of assay and sample replicates reported (1 setup with n=4 replicates) are more representative of preliminary data versus presentation of sufficient data sets for complete analysis. The addition of 4-8 additional replicates (providing 2-3 assay setups) would strengthen the results, assuring that 50% positive cutoff rates (2/4 denoted as positive) are sufficient to deem a blood sample positive.

6. PLOS authors have the option to publish the peer review history of their article (what does this mean?). If published, this will include your full peer review and any attached files.

Reviewer #1: **Yes: **Gianluigi Zanusso

Reviewer #2: No

Reviewer #3: No

---

## [Author Response · Author response to Decision Letter 0]

3 Oct 2023

Response to reviewers

Response to academic editor

Additional Editor Comments: Thank you for submitting your interesting manuscript to PLoS One. Three experts have now reviewed it, and their comments are attached. All of the reviewers indicated that your study is interesting and the subject important. However, two reviewers found that your results so far are a bit preliminary, and that it is important for you to add additional analyses of cases and controls to make this a robust study. One reviewer questioned the inclusion of the PMCA results, but I think that the comparison of assay sensitivities provides some useful context. I will leave up to you to decide what you want to do on that point. Overall, I would ask for a major revision to address the comments of the reviewers (point-by-point). In particular, you should provide additional data to substantiate your findings as requested by Reviewers 2 & 3.

We thank the academic editor for taking the time to review our manuscript, and for his encouraging comments. We have taken note of the comments that some of the results were rather preliminary and have performed additional experiments to substantiate our findings. Specifically, we have performed additional WB IOME RT-QuIC “proof of principle” experiments testing a larger panel of negative controls, and ensuring that all available blood samples were tested in at least three independent experiments (Fig 5, Table 2). 

Although reviewer 2 questioned the inclusion of PMCA results in the original draft of the manuscript, we agree with the academic editor that the inclusion of this data provides useful context to the study. We believe the comparison between assay sensitivities adds value to the manuscript and will potentially be useful for other researchers. Therefore, we have chosen to keep the PMCA results in the revised manuscript.

Response to reviewer 1

Reviewer #1: This study is of high scientific value, it has been technically well performed and the flow of the experiments is appropriate. The paper is well written. Although in animals BSE strain might be presumably still circulating and a blood test might be applicable, in humans no vCJD cases are reported since 10 years, except those incidentally infected. In addition, the estimate of vCJD cases infected by transfusion was initially high but then not confirmed. In summary, this is a high quality study, carried out by top level prion scientists but with an unfortunately weakened impact to public health by the disappearance of BSE.

We thank the reviewer for taking the time to review our manuscript, and for their generous and encouraging comments. Although there have been no new cases of vCJD in recent years, secondary transmission still poses a credible threat to public health. Therefore, we believe that there is still a pressing need for a reliable blood test for BSE/vCJD that is amenable to high-throughput/screening applications. Moreover, we would argue that there is a more general requirement for new pre-treatment strategies to enable prion detection in blood (which is readily accessible, but notoriously difficult to work with). The methodology reported in this study may be useful for early detection of prionemia in blood samples from other animal/human prion diseases.

Response to reviewer 2

Reviewer #2: Thomas et al describe the use of iron oxide beads followed by RT-QuIC to detect ovine BSE prion seeding activity in ovine whole blood (WB RT-QuIC). This manuscript is well written and describes a promising, albeit very preliminary, approach to the detection of prions in blood.

We thank the reviewer for taking the time to review our manuscript, and for their valuable comments and feedback. We agree that some of the findings presented in the original draft of the manuscript were a bit preliminary. Therefore, we have performed additional experiments to substantiate our findings, including additional analysis of cases and controls. We hope that the additional data presented in the revised manuscript will satisfy any concerns surrounding the robustness of the study (see specific comments below).

Major concerns: It is this reviewer’s opinion that, although the findings described in this manuscript are promising, they are still very preliminary. This study only includes a very small number of samples, with only 3 negative control endogenous ovine blood samples in Fig. 4 and Table 2, likely only tested once. Additional testing should be done.

We feel that there may have been some confusion surrounding the two WB IOME RT-QuIC datasets presented in the original manuscript, as Fig 4 and Table 2 represent two different experiments.

In the original manuscript, Fig 4 describes initial optimisation of the WB IOME RT-QuIC assay (to determine analytical parameters of the assay, specifically the 20 h cut-off time). It is a representative dataset, in which we tested 8 prion positive samples (including a dilution series of one sample, M200) and 10 negative control samples (negative controls were combined and plotted as a mean). This experiment was performed once, with n=4 replicate reactions/sample (equivalent to a total of n=32 prion positive reactions and n=40 negative control reactions), which we feel is sufficient for a representative optimisation experiment which was designed solely to illustrate how we determined cut-off. Amyloid formation rates for individual replicate reactions (including those from the 10 negative controls) were originally included in a separate supplementary figure (S4 Fig); we acknowledge that this may have been confusing. We have now merged the data into a single figure (Fig 4 in the revised manuscript), in the hope that this makes things clearer for the reader. 

In the original manuscript, Table 2 and Fig 5 describe the same “proof of principle” experiment, in which the analytical parameters of the WB IOME RT-QuIC assay had already been pre-determined. Negative controls for this experiment were included in a separate supplementary figure (S6 Fig). The reviewer is correct in saying that this was a single experiment that tested only 3 negative control samples. Therefore, we have performed additional experiments testing a larger panel of negative controls, ensuring that each blood sample was tested in at least three independent experiments (apart from one negative control sample, N173 10 mpi, which had limited availability) to substantiate our findings.

• Two repeat experiments were performed, in which the original panel of 12 test samples and 3 negative controls was tested, plus an additional 7 negative control samples (n=4 replicates/sample).

• A third experiment was performed, in which 9 of the negative control samples were tested (n=4 replicates/sample) – sample N173 10 mpi was omitted, as previously discussed.

Furthermore, all data from this experiment (including all control data) have now been merged into a single figure (Fig 5) and table (Table 2) to avoid any confusion.

This study really needs at the very least testing of an equal number of prion positive and negative samples to confirm the findings and make this a publishable unit. This is the reason this reviewer will request “Major reviews”.

The updated WB IOME RT-QuIC “proof of principle” experiment with additional data (detailed in Fig 5 and Table 2) tested 12 prion positive samples in 3 independent experiments (equivalent to n=144 BSE positive reactions) and 10 negative controls (equivalent to n=128 negative control reactions). When combined with the optimisation experiment in Fig 4, a total of n=20 prion positive samples and n=20 negative control samples have now been tested by WB IOME RT-QuIC. We are hopeful that this is sufficient to make this a publishable unit.

Results from testing of additional negative ovine whole blood samples are reported in Supplementary Figure 4, if these are different animals the data should probably be included in Figure 4.

We agree with the reviewer and have now merged these data into a single figure (Fig 4).

Minor concerns: This reviewer is a little confused by the inclusion of the PMCA data in this study. The results show that the PMCA is more sensitive than the RT-QuIC but the authors choose to test the endogenous samples with the latter assay, presumably because the RT-QuIC is better suited for practical diagnostic applications. Although the PMCA data is interesting, because it was not used to test endogenous whole blood samples, it just seems to be a confusing addition to the manuscript.

Although we respect the views of the reviewer and thank them for their feedback, we have decided to keep the PMCA data in the revised manuscript. As noted by the academic editor, we believe that the inclusion of this data provides useful context to the study. As equivalent buffy coat samples from some animals/time points were previously tested by PMCA (Salamat et al, 2021, Table 2), we believe that comparison between PMCA and RT-QuIC assay sensitivities adds value to the manuscript and may potentially be useful for other researchers.

Furthermore, several of the supplementary figures and tables, which contain key information about the development of the WB RT-QuIC and the in vivo data, could be moved into the main text.

Again, we agree with the comments of the reviewer and we have now moved two supplementary figures (S4 and S6 Figs) into the main text (Figs 4 and 5, respectively), as they contained important information about the development of the WB IOME RT-QuIC assay.

Which negatives were included in Fig. 4 ? Please include samples IDs.

We thank the reviewer for highlighting this oversight. We have now included all negative control sample IDs in Fig 4C.

The PBS spiking experiments should include negative control brain homogenate spiked PBS.

Again, we thank the reviewer for highlighting this oversight. We have performed an additional RT-QuIC experiment testing PBS spiked with negative control sheep brain homogenate (10-2 and 10-3 dilutions). This data has now been added to Fig 3C.

Response to reviewer 3

Reviewer #3: The authors modified prion amplification assays to develop a haematogenous prion detection assay using a vCJD large animal model (BSE-infected sheep/blood transfusion passage). They provide comparison of modified PMCA and RT-QuIC assays. They demonstrate the utility of mb-PMCA in the detection of blood-borne prions and correlate this to mouse bioassay. Great effort was put forth in recombinant protein optimization, time to positive cut-off, and assay readouts. These components of the work provide great insight into the use of specific recombinant proteins in the detection of prions present in blood. Further exploration of sample pre-treatment is warranted as challenges have been encountered in the development of assays that are able to detect prions during the very early phases of disease, especially detection in blood. The manuscript is exceptionally well written and organized.

We thank the reviewer for taking the time to review our manuscript, and greatly appreciate their generous and encouraging comments.

Major flaw: The number of assay and sample replicates reported (1 setup with n=4 replicates) are more representative of preliminary data versus presentation of sufficient data sets for complete analysis. The addition of 4-8 additional replicates (providing 2-3 assay setups) would strengthen the results, assuring that 50% positive cut-off rates (2/4 denoted as positive) are sufficient to deem a blood sample positive.

We agree with the comments of the reviewer, and acknowledge that some of the findings presented in the original draft of the manuscript were more representative of preliminary data. The reviewer is correct in saying that the original WB IOME RT-QuIC “proof of principle” experiment represented a single experimental setup with n=4 replicates, and we agree that additional testing was required to strengthen our results. Therefore, we have performed three repeat experiments to substantiate our findings, ensuring that each blood sample was tested in at least three independent experiments, representing an additional 8-12 replicates per sample (apart from negative control sample, N173 10 mpi, which had limited availability):

• Two repeat experiments were performed, in which the original panel of 12 test samples and 3 negative controls was tested, plus an additional 7 negative control samples (n=4 replicates/sample).

• A third experiment was performed, in which 9 of the negative control samples were tested (n=4 replicates/sample) – sample N173 10 mpi was omitted, as previously discussed.

These data support our previously determined analytical parameters for WB IOME RT-QuIC (re: cut-off time and the definition of a positive sample), and we trust that the resulting dataset (Fig 5, Table 2) is now robust enough to warrant publication.

Journal Requirements

We have changed formatting and file naming to the specifications required by PLOS ONE. All figures have been checked by PACE.

We have edited the Materials and Methods section to include this information, see sub-sections “Sheep experiments (source of blood samples)” and “Mouse experiments (tgOvARQ bioassay).

The original uncropped and unadjusted SDS-PAGE gel images accompanying this study are now included in a supplementary file (S1_raw_images), according to specifications required by PLOS ONE. The original software files are available if required.

We apologise for this oversight. The phrase “data not shown” was used twice in the original manuscript. In the first instance (line 502 in the original manuscript, line 514 in the revised manuscript) we have provided data to support our statement in supplementary information (S4 Fig). In the second instance (line 676 in the original manuscript), the phrase referring to these data was removed.

---

## [Decision Letter · Decision Letter 1]

19 Oct 2023

Development of a sensitive real-time quaking-induced conversion (RT-QuIC) assay for application in prion-infected blood.

PONE-D-23-21489R1

Dear Dr. Thomas,

We’re pleased to inform you that your manuscript has been judged scientifically suitable for publication and will be formally accepted for publication once it meets all outstanding technical requirements.

Kind regards,

Byron Caughey

Academic Editor

PLOS ONE

Additional Editor Comments (optional):

Reviewers' comments:

Reviewer's Responses to Questions

**Comments to the Author**

1. If the authors have adequately addressed your comments raised in a previous round of review and you feel that this manuscript is now acceptable for publication, you may indicate that here to bypass the “Comments to the Author” section, enter your conflict of interest statement in the “Confidential to Editor” section, and submit your "Accept" recommendation.

Reviewer #2: All comments have been addressed

2. Is the manuscript technically sound, and do the data support the conclusions?

Reviewer #2: Yes

3. Has the statistical analysis been performed appropriately and rigorously? 

Reviewer #2: Yes

4. Have the authors made all data underlying the findings in their manuscript fully available?

Reviewer #2: Yes

5. Is the manuscript presented in an intelligible fashion and written in standard English?

Reviewer #2: Yes

6. Review Comments to the Author

Reviewer #2: (No Response)

7. PLOS authors have the option to publish the peer review history of their article (what does this mean?). If published, this will include your full peer review and any attached files.

Reviewer #2: No

---

## [Editor Report · Acceptance letter]

24 Oct 2023

PONE-D-23-21489R1 

Development of a sensitive real-time quaking-induced conversion (RT-QuIC) assay for application in prion-infected blood. 

Dear Dr. Thomas:

I'm pleased to inform you that your manuscript has been deemed suitable for publication in PLOS ONE. Congratulations! Your manuscript is now with our production department. 

Kind regards, 

on behalf of

Dr. Byron Caughey 

Academic Editor

PLOS ONE